# Membrane-embedded polar residues target membrane proteins for degradation by the quality control protease FtsH

Michal Chai-Danino [1], Noy Ravensary-Modin[1], Vasiliy I. Vladimirov [1], Tetiana Onyshchuk[1], Martin Plöhn [2], Alon B. D. Barshap [1], Aseel Bsoul[1], Hadas Peled-Zehavi [1] & Nir Fluman [1] ✉

The biogenesis of membrane proteins (MPs) is inherently error-prone, and is therefore monitored by quality control mechanisms that remove faulty MPs. A key challenge for this surveillance is to recognize misfolded MPs, but how this is achieved remains poorly understood. Here we reveal how FtsH, the main MP quality control protease in *Escherichia coli*, specifically targets faulty MPs. By analyzing the in vivo degradation of two substrates, we show that lipid-facing polar residues trigger FtsH-mediated degradation. In folded MPs, such polar residues are usually buried in the protein core. Their exposure to the membrane can therefore signal misfolding and promote degradation. Strikingly, lipid-facing polar residues can even trigger degradation of a folded protein, and do not require the extended cytosolic regions typically needed for other FtsH substrates. Recognition depends on the FtsH transmembrane domain and on specific polar residues within it. Thus, sensing misfolding within the membrane helps maintain the integrity of the membrane proteome.

Membrane proteins (MPs), which make up a quarter of every proteome[1], are continuously synthesized yet prone to misfolding[2–5]. In cells, misfolded and unassembled proteins can be toxic, and must be eliminated by protein quality control (QC) systems to maintain homeostasis[6–8]. However, while QC pathways for soluble proteins are well characterized, the mechanisms that govern MP QC remain less understood.

QC hinges on *selective* degradation, requiring precise discrimination between misfolded MPs and folded ones. This selectivity is essential to avoid the risk of eliminating folded functional proteins, which may compromise fitness[9,10]. Yet the mechanisms underlying the recognition of misfolded MPs are only beginning to be revealed. Several features of misfolded MPs have emerged as cues for QC recognition. These include features outside the membrane, such as misfolded extramembrane domains[11,12], or transmembrane (TM) segments that fail to properly integrate into the bilayer[13,14]. Other crucial cues are embedded within the membrane, such as the presence of unstable helices prone to local unwinding[15–21], or polar residues that become exposed to the hydrophobic core of the membrane[22–26].

Several studies suggest that when MPs misfold, their transmembrane helices can partially unpack or separate from one another[27–29], potentially exposing previously buried residues to the surrounding lipids. Among these, polar and charged residues are particularly informative. These residues are often found within transmembrane regions, where they support essential functions such as ion coordination, substrate binding, folding, and oligomerization. However, they are typically buried within the core of folded MPs, while the lipid-exposed surface remains highly hydrophobic (Fig. 1a). TM rearrangements induced by misfolding may therefore expose polar residues to the membrane environment, providing a signal for misfolding (Fig. 1b). Indeed, such membrane-embedded polar residues have been implicated as QC cues in several systems[23,24,30–33]. Yet how they are recognized, and whether this occurs directly or requires accessory factors, remains unclear. Moreover, these principles have been established

[1]Department of Biomolecular Sciences, Weizmann Institute of Science, Rehovot, Israel. [2]Department of Biochemistry and Biophysics, Stockholm University, Stockholm, Sweden. ✉e-mail: nir.fluman@weizmann.ac.il

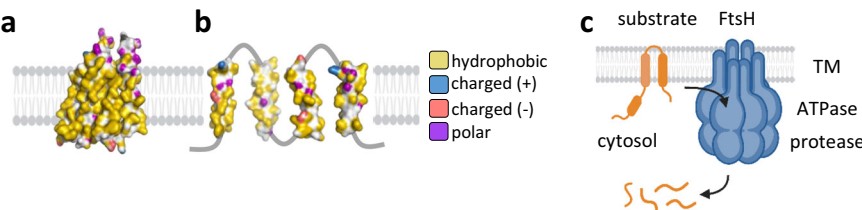

**Fig. 1 | Structural and cellular context of FtsH-mediated MP degradation. a, b** Hydrophobic, polar, and charged groups in the predicted structure of MdtJ, a prototypic MP with four TMs. Hydrophobic carbons that do not lie next to electronegative atoms are colored yellow, polar groups in Ser, Thr, Asn, Gln, and His are colored purple. Charged groups in Asp, Glu, Lys, and Arg are colored red or blue, depending on the charge. The membrane was created using BioRender. Kalinin, I. (2026) https://BioRender.com/50mmo62. **a** AlphaFold2 prediction of MdtJ structure, showing the side that faces the lipids in the folded dimer. The lipid-facing surface is hydrophobic, as is typical for a folded MP. **b** Membrane-embedded polar residues in the individual TMs of MdtJ, representing a hypothetical fully unfolded state. While the model is likely not conformationally accurate, it illustrates that many polar residues may potentially face the lipid upon unfolding. **c** Illustration of MP degradation by hexameric FtsH. The substrate is pulled from the membrane by the AAA+ ATPase domain and fed into a proteolytic chamber for processive degradation. Created in BioRender. Kalinin, I. (2026) https://BioRender.com/tsbxmt8.

primarily in eukaryotic contexts, and whether they also apply to bacterial QC systems, such as FtsH, remains unknown.

The FtsH family of proteases is the principal MP QC machinery in bacteria, mitochondria, and chloroplasts[34–38]. FtsH forms hexamers composed of two transmembrane helices (TMs) per subunit and a large soluble domain localized to the cytosol in bacteria[37,39]. The soluble domain is structurally resolved and harbors an AAA+ ATPase that can pull proteins from the membrane and processively feed them into a proteolytic chamber within the hexamer (Fig. 1c)[40–43]. By contrast, the structure of the membrane domain is poorly resolved, and its function beyond serving as a membrane anchor remains unclear[41,44–47].

Despite decades of study, how FtsH proteases discriminate misfolded from folded MPs remains unknown. Early studies utilizing soluble substrates suggested that the *E. coli* FtsH does not possess a robust unfoldase activity[48,49], implying that folded proteins would be spared, thus allowing it to differentiate between folded and unfolded proteins. However, later studies with MP substrates revealed that FtsH can, in fact, unfold MPs[50]. Substrate recognition has also been linked to specific sequence features, including degrons like ssrA and others[51–53], and long cytosolic polypeptide stretches that provide handles for the ATPase to initiate pulling[54,55]. Additionally, certain adaptors can direct substrates to FtsH, likely irrespective of their folding state[15,56]. Yet none of these mechanisms explains how misfolded MPs are recognized while folded MPs are spared.

Here, we address these knowledge gaps by examining how FtsH recognizes aberrant MPs. Using a combination of engineered model substrates, we show that polar residues facing the lipid bilayer are both sufficient and essential for FtsH-dependent degradation. Remarkably, this degradation occurs in the absence of extended cytosolic degrons, indicating that substrate recognition can take place entirely within the membrane. Furthermore, we find that recognition depends on the transmembrane domain of FtsH itself, which is structurally adapted to sense aberrant membrane features. These findings reveal a membrane-embedded recognition mechanism that expands our understanding of MP QC specificity.

## Results

### Lipid-facing polar residues can target a folded protein for degradation by FtsH

To examine whether lipid-facing polar residues can target a membrane protein for FtsH-mediated degradation, we utilized *Yersinia frederiksenii* ASBT, a bacterial homolog of the human apical sodium bile acid transporter. The structure of ASBT reveals a predominantly hydrophobic surface exposed to the lipid bilayer[57] (Fig. 2a left). We engineered it to present a polar glutamine at position 73, in the middle of TM3, replacing a lipid-facing leucine (Fig. 2a, right). TM3 was selected since it is the most hydrophobic TM in ASBT, and is expected to

robustly insert into the membrane upon mutation, avoiding misfolding due to problems in TM insertion (Supplementary Fig. 1a). The mutated ASBT would thus atypically present a polar residue to the lipid bilayer, despite being folded.

We first confirmed that ASBT-L73Q is properly folded. Wild-type ASBT and its L73Q mutant were purified from FtsH-deleted cells, and their thermal denaturation was measured by differential scanning fluorimetry. The mutant and wild-type displayed an unfolding transition at ~65 °C, suggesting similar thermal stabilities (Fig. 2b). As a negative control, we examined an ASBT mutant lacking its centrally-located TM10, which is unlikely to stably fold without the many tertiary interactions provided by this TM (Supplementary Fig 1b). Indeed, the ΔTM10 mutant showed no indication of thermal unfolding, confirming that it is unfolded already at low temperatures (Fig. 2b). Thus, ASBT-L73Q appears to be stably folded when purified in detergent.

Having established the stability of ASBT-L73Q in detergent, we further probed its folding in native membranes using a crosslinking assay. The assay tests the structural proximity of two residues that would only be within crosslinking distance if the protein is folded. To this end, cysteines were introduced into otherwise cysteine-less ASBT at two positions, L92C and V219C, which lie close in the three-dimensional structure of folded ASBT (Supplementary Fig. 1c). Several observations confirm that the two cysteines are effectively crosslinked by dibromobimane (dBBr) (Supplementary Fig. 1d). First, the intra-molecularly crosslinked protein displayed a shift in gel migration (Fig. 2c, lane 2, Coomassie). Secondly, dBBr becomes fluorescent once it reacts with two cysteines simultaneously[58], and indeed, the ASBT-dBBr adduct is highly fluorescent (Fig. 2c, bottom). Thirdly, neither gel-shift nor increased fluorescence was observed in ASBT mutants harboring only a single Cys at position L92C or V219C, confirming that both cysteines crosslink together (Supplementary Fig. 1e). ASBT-L73Q was fully crosslinked in native membranes, indicating that residues L92C and V219C are proximal in the mutant, as in the wild-type, while the misfolding caused by truncation of TM10 abolished crosslinking (Fig. 2c). Considering the location of the crosslinking sites (Supplementary Fig. 1c, d, see legend), these results confirm that the tertiary structure of ASBT-L73Q, especially related to TM3,4, and 9, is maintained in the native membrane. While we cannot exclude a local perturbation to the structure, our results suggest that the L73Q mutant retains its overall structure and stability.

The folded state of ASBT-L73Q suggests it should be spared degradative QC. Remarkably, however, this mutant gets rapidly recognized and degraded when expressed in *E. coli*, as if it were an aberrant protein (Fig. 2d, e). Interestingly, nearly 40% of ASBT-L73Q escapes degradation by an unknown mechanism that deserves further research. Knockout of the *FtsH* gene significantly decreased the degradation rate, but did not abolish it (Fig. 2d bottom, Fig. 2e right

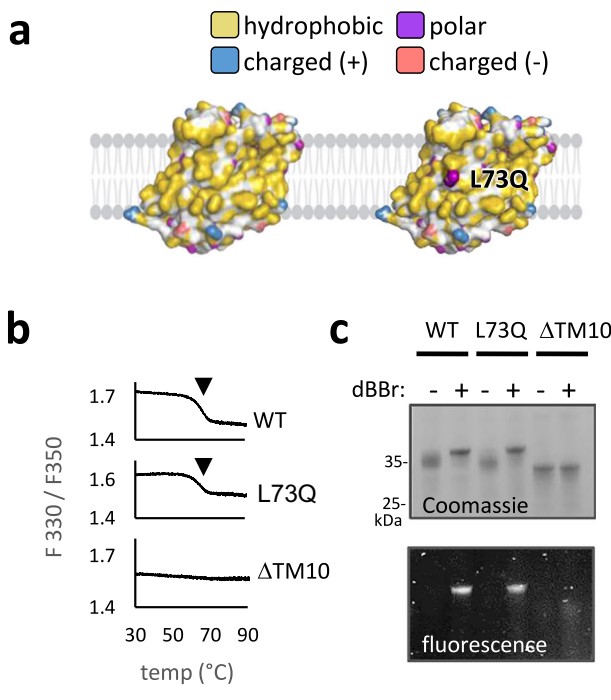

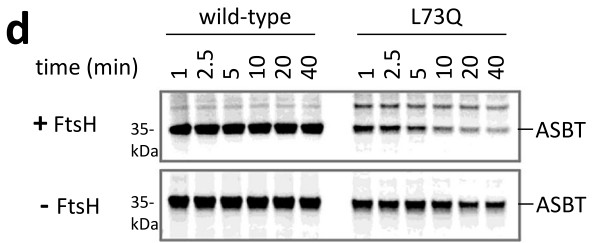

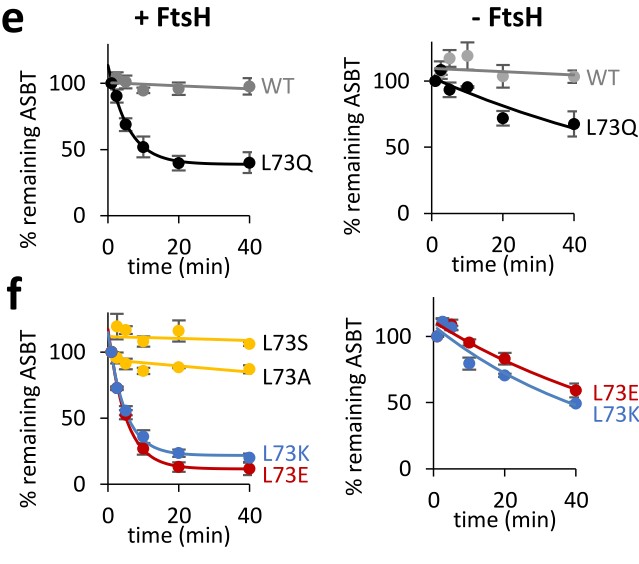

**Fig. 2 | A lipid-exposed polar residue directs folded ASBT for degradation. a** The hydrophobic surface presented to the membrane by ASBT, and the hydrophilic group of glutamine pointing to the membrane in the L73Q mutant. Color coding is shown in Fig. 1a, b. The membrane was created using BioRender. Kalinin, I. (2026) https://BioRender.com/50mmo62. **b** Thermal denaturation suggests that ASBT-L73Q is stably folded. Purified ASBT, its folded L73Q mutant, or misfolded ΔTM10 mutant were gradually heated, and denaturation was followed by the ratio of fluorescence at 330 and 350 nm. The downward arrow indicates the unfolding transition. Representative data from two biological repeats are shown. **c** Crosslinking of ASBT L92C-V219C by dBBr suggests that ASBT-L73Q is folded. Following crosslinking in native *E. coli* membranes, the proteins were purified, run on SDS-PAGE, and detected by Coomassie and in-gel fluorescence. Crosslinking is shown by a shift in gel migration and the appearance of a fluorescent band. Representative data from three biological repeats are shown. **d** Degradation of ASBT and its L73Q mutant, followed by radioactive pulse-chase and auto-radiography. L73Q is degraded in an FtsH-dependent manner. **e, f** Quantifications of the degradation of ASBT and its mutants in the presence and absence of FtsH. Shown are means ± SEM, quantified from three biological repeats (as in (d)). Solid lines depict nonlinear regression fits to an exponential decay equation, in which a fraction of the protein remains stable over time. **e** The degradation of ASBT-L73Q is dramatically slowed down in the absence of FtsH, suggesting that FtsH degrades ASBT-L73Q. **f** FtsH degrades the charged L73E and L73K mutants, but not the less polar L73A and L73S mutants. Source data are provided as a Source Data file.

uncharged glutamine. Like the L73Q mutant, the absence of FtsH did not completely abolish the degradation of the charged mutants, suggesting that an additional redundant protease(s) exists (Fig. 2f, right). These findings demonstrate that a single lipid-facing polar residue is sufficient for targeting even a folded MP for degradation, highlighting the sensitivity of the QC system to polar residues.

**Polar residues are essential for orphan MdtJ degradation by FtsH**

The glutamine mutant of ASBT departs from a natural QC substrate and exhibits abnormal membrane polarity despite being well folded. To investigate how FtsH recognizes a more naturally relevant misfolded MP, we focused on the family of dimeric Small Multidrug Transporters (SMR). These proteins provide an ideal model, as they remain unfolded until they dimerize, allowing us to control their folding status[27]. In the monomeric unfolded state, their TMs are at least partially unpacked[27]. In addition, FtsH was shown to degrade orphan monomers of an SMR family member[59,60]. To manipulate the dimerization and folding of an SMR protein in a cellular context, we turned to heterodimeric family members, allowing the expression of one subunit without its partner[61]. Indeed, when we expressed the *Lawsonia intracellularis* MdtJ (UniProt accession, Q1MPU8) as an orphan GFP-MdtJ fusion, it was degraded in an FtsH-dependent manner, with a half-life of about 40 min (Fig. 3a, b, e, f). Co-expression of its cognate dimerization partner, MdtI (UniProt accession Q1MPU9), abolished the degradation (Fig. 3a, b). Thus, FtsH degrades only orphan and aberrant MdtJ, consistent with QC-targeted selectivity.

We hypothesized that polar residues within the transmembrane regions of MdtJ may target the misfolded protein for degradation. The MdtJI dimer has no solved structure, yet its high-confidence AlphaFold prediction is consistent with the solved structures of dimers from the SMR family[61]. According to the predicted structure, the folded and assembled MdtJ has 4 TMs and exposes a hydrophobic surface to the membrane (Fig. 1a). In the misfolded state, these helices may fully or partially unpack from each other, potentially exposing polar residues to the membrane (Fig. 3c). We mutated each of its transmembrane polar residues, alone or in adjacent pairs, and substituted them with hydrophobic residues of similar size. Most of these residues were dispensable for FtsH-mediated degradation (Fig. 3d). However, two substitutions, S56A and E77L, significantly slowed down the degradation when mutated alone, and almost completely abolished the FtsH-mediated degradation when mutated together (Fig. 3d–f). These results are consistent with the notion that the TMs of MdtJ at least

panel). Thus, while FtsH appears to be the major protease degrading ASBT-L73Q, membrane-facing polar residues may serve as a general signal for other *E. coli* proteases. Additional substitutions in position 73 showed that the polarity of the residue correlated with degradation (Fig. 2f, left). Relatively apolar substitutions at this position did not lead to degradation, whereas charged residues, whether negative or positive, elicited even faster and more complete degradation than the

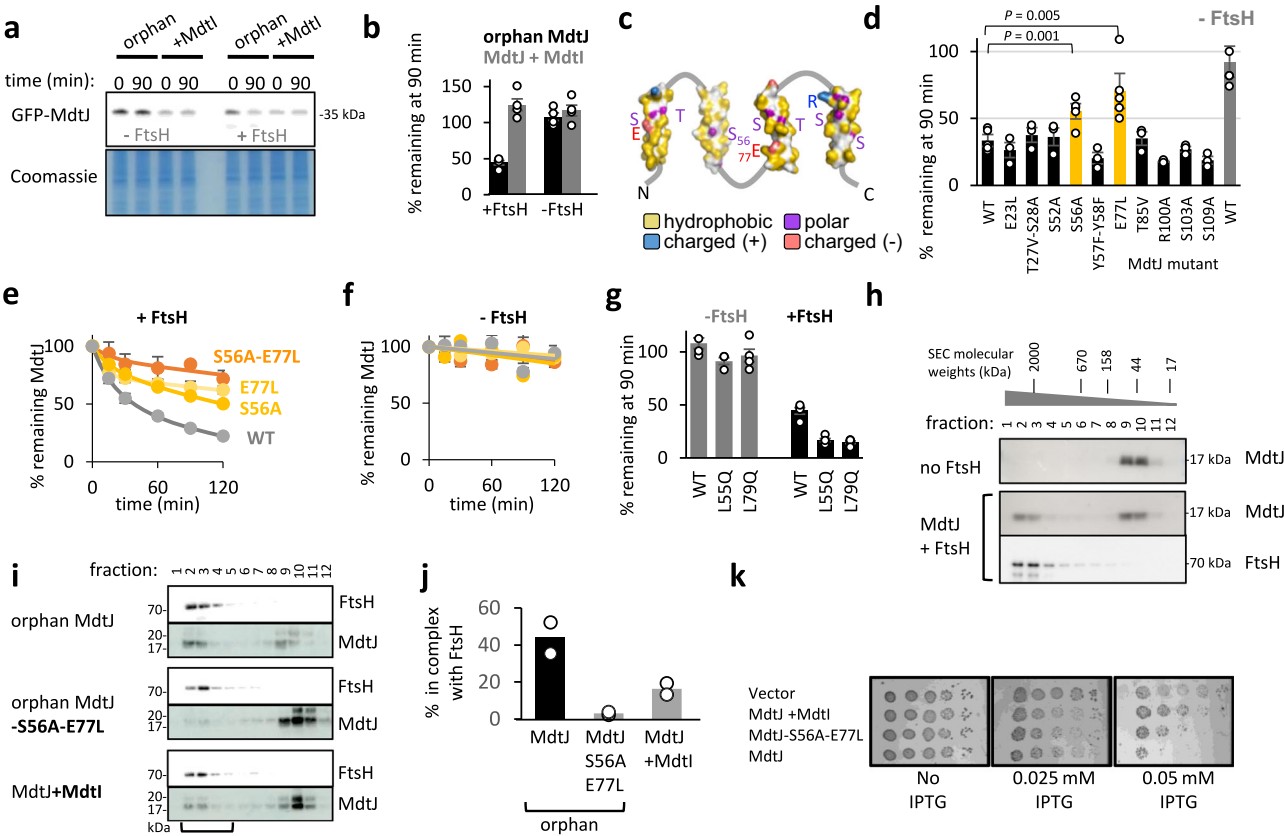

**Fig. 3 | Polar residues mediate the degradation of orphan MdtJ by FtsH. a** FtsH mediates degradation of orphan GFP-MdtJ. MdtJ was expressed either as an orphan protein or with its cognate partner MdtI. At time 0, spectinomycin was added to stop translation, and the degradation was assessed after 90 min. In the FtsH knockout strain, there is no evident degradation, while co-expressed FtsH conferred degradation of orphan MdtJ. Coomassie staining is shown as a loading control. **b** Quantification of the experiment shown in (**a**). Bars show averages ± SEM of three biological replicates. **c** Membrane-embedded polar residues in the individual TMs of MdtJ, which may potentially be recognized by FtsH. S56 and E77 are indicated. **d** Effect of substituting MdtJ polar residues with similarly sized apolar amino acids on degradation, as assessed by the amount of orphan GFP-MdtJ remaining after 90 min. Mutations to S56 and E77 perturb the degradation. Degradation of wild-type (WT) GFP-MdtJ in the presence or absence of FtsH is shown as a control. Bars show averages ± SEM, from at least three biological replicates (see Source Data file for exact n). *P* values are taken from a two-sided *t*-test. Raw data are presented in Supplementary Fig. 2a. **e** Degradation time course of GFP-MdtJ mutants in the presence of FtsH, showing that the mutations E77L or S56A slow down the degradation, and their combination almost completely abolishes degradation. Shown are averages ± SEM, as quantified from three biological replicates similar to Fig. 2b. Solid line: the data were fit to a double-exponential equation $Y(t) = A_1 \cdot e^{-k_1 \cdot t} + A_2 \cdot e^{-k_2 \cdot t}$, where $A_{1,2}$ are protein fractions and $k_{1,2}$ are rate constants.

Raw data are presented in Supplementary Fig. 2b. **f** Same as (**e**), but in the absence of FtsH. Raw data are presented in Supplementary Fig. 2b. **g** Effect of L55Q and L79Q mutations on GFP-MdtJ degradation in the absence (gray) and presence (black) of FtsH. Bars show averages ± SEM of at least three biological replicates (see Source Data file for exact n). Raw data are shown in Supplementary Fig. 2f. **h** Interaction of MdtJ and FtsH. MdtJ membranes were mixed with membranes containing no FtsH (top) or membranes from cells overexpressing FtsH. The solubilized complexes were run on size exclusion chromatography to analyze their size. The elution fractions of molecular weight standards are shown on top. In the presence of FtsH, MdtJ co-eluted with FtsH fractions, suggesting they form a complex. Representative data from two biological repeats are shown. **i** The mutation of S56A-E77L, as well as the co-expression of MdtI, diminishes the interaction of FtsH with MdtJ, as analyzed by size exclusion chromatography, similar to (**h**). **j** Average values of the percent of MdtJ found in a high molecular weight complex, as determined from densitometry of two repetitions of the experiment presented in (**i**). **k** Effect of polar residues of MdtJ on *E. coli* growth. A 10-fold serial dilution of the bacterial culture was spotted onto LB-agar plates under no induction (no IPTG) or under increasing IPTG-controlled induction. The level of toxicity is inferred by reduced growth compared to cells harboring an empty vector. Source data are provided as a Source Data file.

partially unpack in the unfolded monomeric state[27], since S56 is predicted to be buried in the core of the folded monomer and would not be accessible for recognition. Interestingly, while S56 and E77 were crucial, other, equally polar residues were dispensable for degradation. This suggests that the positional or structural context of the polar residues plays a role in their recognition, which deserves further study. Nevertheless, the role of S56 and E77 implies that membrane-embedded polar residues are essential for MdtJ recognition by FtsH.

Since mutating polar residues inhibited degradation, we hypothesized that introducing an additional polar residue might accelerate MdtJ degradation. Glutamines were introduced at two positions, L55Q and L79Q, positioned deep within the membrane and lying close in sequence to S56 and E77 that mediate orphan MdtJ degradation

(Supplementary Fig. 2c, d). Notably, TMs 2 and 3, which harbor the mutations, are sufficiently hydrophobic to insert into the membrane even with the polar mutations (Supplementary Fig. 2c). Indeed, both mutations accelerated the degradation of monomeric MdtJ by FtsH (Fig. 3g), further supporting that membrane-embedded polar residues direct orphan MdtJ for degradation by FtsH.

We next biochemically characterized the interaction between FtsH and MdtJ. Membranes from cells expressing MdtJ were mixed with membranes from cells expressing FtsH by freeze-thaw cycles, to allow the proteins to bind. The complexes formed were solubilized by detergent and analyzed by size exclusion chromatography. When MdtJ was mixed with membranes lacking FtsH, it eluted as a particle of about 100 kDa, consistent with a ~ 15 kDa protein embedded in a

detergent micelle (Fig. 3h). By contrast, when mixed with membranes expressing FtsH, nearly half of it eluted as a large, ~2 MDa complex. This large size coincided with the apparent size of the major FtsH complex, likely corresponding to either 12- or 24-mer FtsH complexes previously observed, together with lipids and detergent (Fig. 3h)[43,44,62]. The FtsH-MdtJ interaction was significantly reduced when MdtJ was co-expressed with its partner MdtI, and was diminished to less than 5% when MdtJ was mutated at the two critical polar residues, S56A-E77L (Fig. 3i, j). This suggests that FtsH selectively binds orphan MdtJ, and can distinguish it from its mutant form or from the MdtJI complex. Thus, polar residues promote FtsH binding, and their masking, either by mutation or by dimerization with the cognate partner, disrupts this interaction.

Orphan MdtJ was toxic to cells, and interestingly, this effect depended in part on its polar residues. When expressing IPTG-controlled orphan MdtJ, it exhibited toxicity and inhibited growth. The toxicity was alleviated by co-expression of its dimerization partner, MdtI (Fig. 3k), suggesting that misfolded orphan MdtJ is harmful to the cell. The S56A-E77A double mutant showed reduced toxicity despite remaining unfolded and escaping degradation by the cellular QC. Thus, lipid-facing polar residues not only mediate degradation but also contribute to toxicity, underscoring the importance of eliminating proteins that possess them.

## Substrate membrane determinants are sufficient for FtsH-mediated degradation

Previous studies showed that FtsH-mediated degradation of membrane proteins typically requires an unstructured cytosolic loop or tail of approximately 10–20 amino acids, typically at the N- or C-terminus[50,54,63]. Such regions, referred to as degradation markers, are thought to serve as pulling handles, enabling the cytosolic ATPase domain of FtsH to initiate pulling on the substrate. To determine whether a cytosolic stretch is required for the degradation of MPs harboring membrane-exposed polar residues, we investigated our two model substrates, ASBT and GFP-MdtJ.

Our ASBT construct possesses an N-terminal FLAG-His tag, amounting to cytosolic N- and C-terminal tails of 31 and 13 amino acids, respectively. Since the N-terminal tail length is longer and better matches the previously identified degradation markers, we generated a truncation mutant lacking this region, termed ΔN (Fig. 4a). Notably, the truncation diminished the expression of both wild-type and mutant ASBT-L73Q (Supplementary Fig. 3a), possibly because it affected the translation initiation region[64]. The ΔN-ASBT-L73Q was still degraded in an FtsH-dependent manner, though it was degraded to a lesser extent than the intact ASBT-L73Q (Supplementary Fig. 3a, b). Interestingly, while the intact ASBT-L73Q exhibited partial degradation even in ΔftsH cells, the ΔN mutant was completely dependent on FtsH for degradation (Fig. 4b, c). Thus, FtsH does not require the cytosolic N-terminus to degrade ASBT-L73Q.

We next examined the cytosolic loops and tails of GFP-MdtJ. GFP-MdtJ harbors a ~ 22 amino acid unstructured cytosolic stretch, formed by the unstructured GFP C-terminus and MdtJ's N-terminus, as well as an 11-residue cytosolic C-terminal tail (Fig. 4d). The cytosolic loop between helix 2 and 3 is only 5 residues long and therefore too short to serve as a degradation marker. Shortening the linker (from 22 to 5 residues, Δlinker) slowed down the degradation (Fig. 4d, e). However, an additional deletion in the MdtJ C-terminal tail was degraded as efficiently as the wild-type GFP-MdtJ (Δlinker-ΔC, Fig. 4d, e). These observations suggest that the degradation of GFP-MdtJ does not depend on unstructured cytosolic stretches. However, given that the GFP variant used, sfGFP, requires several minutes to fold, its unstructured pre-folded polypeptide could act as a transient degradation marker. To exclude this possibility, we evaluated the degradation of MdtJ constructs lacking sfGFP and the N- and C-terminal tails using a radioactive pulse-chase assay. While shortening the N- and C-terminal tails slowed down the degradation of MdtJ to some extent, all mutants were degraded in an FtsH-dependent manner (Fig. 4f, g). These results indicate that FtsH can degrade membrane proteins devoid of long unstructured cytoplasmic stretches that serve as degradation markers. This expands recent studies on soluble FtsH substrates, which suggested that extended unstructured loops are not always essential for degradation[65,66].

The degradation of N and C-tail-less MdtJ suggests that FtsH employs an alternative mechanism, where the recognition of membrane-embedded polar residues may replace the reliance on unstructured cytosolic tails. One possible biological outcome is that FtsH may degrade membrane proteins with periplasmic N- and C-termini, devoid of an extended cytosolic loop. To explore this possibility, we tested the degradation of orphan MdtI, a subunit of the MdtJI heterodimer with such a topology. Notably, the cytosolic loops of MdtI are only 8 and 3 residues long, neither of which is sufficiently long to serve as a degradation marker for FtsH (Fig. 4h). Nevertheless, orphan MdtI was degraded in an FtsH-dependent manner (Fig. 4i). FtsH's ability to engage proteins lacking extensive cytosolic regions suggests it can degrade a wider range of substrates than previously recognized. Approximately 21% of the multipass membrane proteins have minimal ( < 20 aa) or absent cytosolic extensions (Fig. 4j). FtsH's adaptability to such diverse topological configurations supports its role as a versatile quality-control protease.

## The FtsH transmembrane domain plays a role in MP QC

We next asked how FtsH recognizes substrates bearing membrane-facing polar residues. FtsH's ability to degrade substrates exhibiting only membrane determinants suggests that the recognition occurs within the membrane, possibly involving its transmembrane domain. FtsH possesses two TMs that anchor it to the membrane and facilitate hexamerization[47]. However, examining their role in substrate recognition yielded conflicting results in different organisms. In *S. cerevisiae*, the TMs appear important for the degradation of some integral MPs[41,46]. In contrast, the TMs of the *E. coli* FtsH appear dispensable, as swapping them with unrelated TMs from the LacY protein retains the degradation of the orphan MP SecY. We speculated that FtsH may use distinct modes of recognition for different MP substrates, and set out to test the role of the TMs in recognizing membrane-facing polar residues.

We first repeated the approach of Akiyama and Ito, swapping the N-terminal, TM-containing part of FtsH with TMs 1 and 2 of LacY[47] (Fig. 5a, b). Two LacY-FtsH chimeras were constructed, one with the wild-type LacY TMs and another, termed *hydrophobic* LacY-FtsH, in which most polar residues in the LacY TMs were replaced by apolar ones (Fig. 5b, legend). To assess whether these chimeras maintained the active structure and ATP-dependent protease activity, we examined their ability to undergo self-trimming. This ATP-dependent process involves autolytic cleavage of the FtsH C-terminus, without impairing the function of the protein[67]. Indeed, when expressed with a 3xHA tag, LacY chimeras exhibited C-terminal self-trimming at rates comparable to wild-type FtsH, indicating that both variants assembled into active hexamers (Fig. 5c and Supplementary Fig. 4a, b). Furthermore, both chimeras efficiently degraded a soluble cytosolic substrate, the λ phage regulatory protein CII[68] (Fig. 5e). However, neither chimera degraded orphan MdtJ (Fig. 5c, upper panel, and Fig. 5d), suggesting that the native transmembrane domain is required for specificity towards MPs with lipid-facing polar residues. This may explain why the LacY−FtsH chimera could not fully complement an ftsH deletion, despite retaining activity against several substrates[47].

## TM1 polar residues play a role in substrate recognition

We next investigated which features of FtsH TMs contribute to MdtJ degradation. The FtsH transmembrane domain is poorly resolved in the current structures[43,62], suggesting that this part of the protein is dynamic. Consistently, an AlphaFold model of the hexameric FtsH had

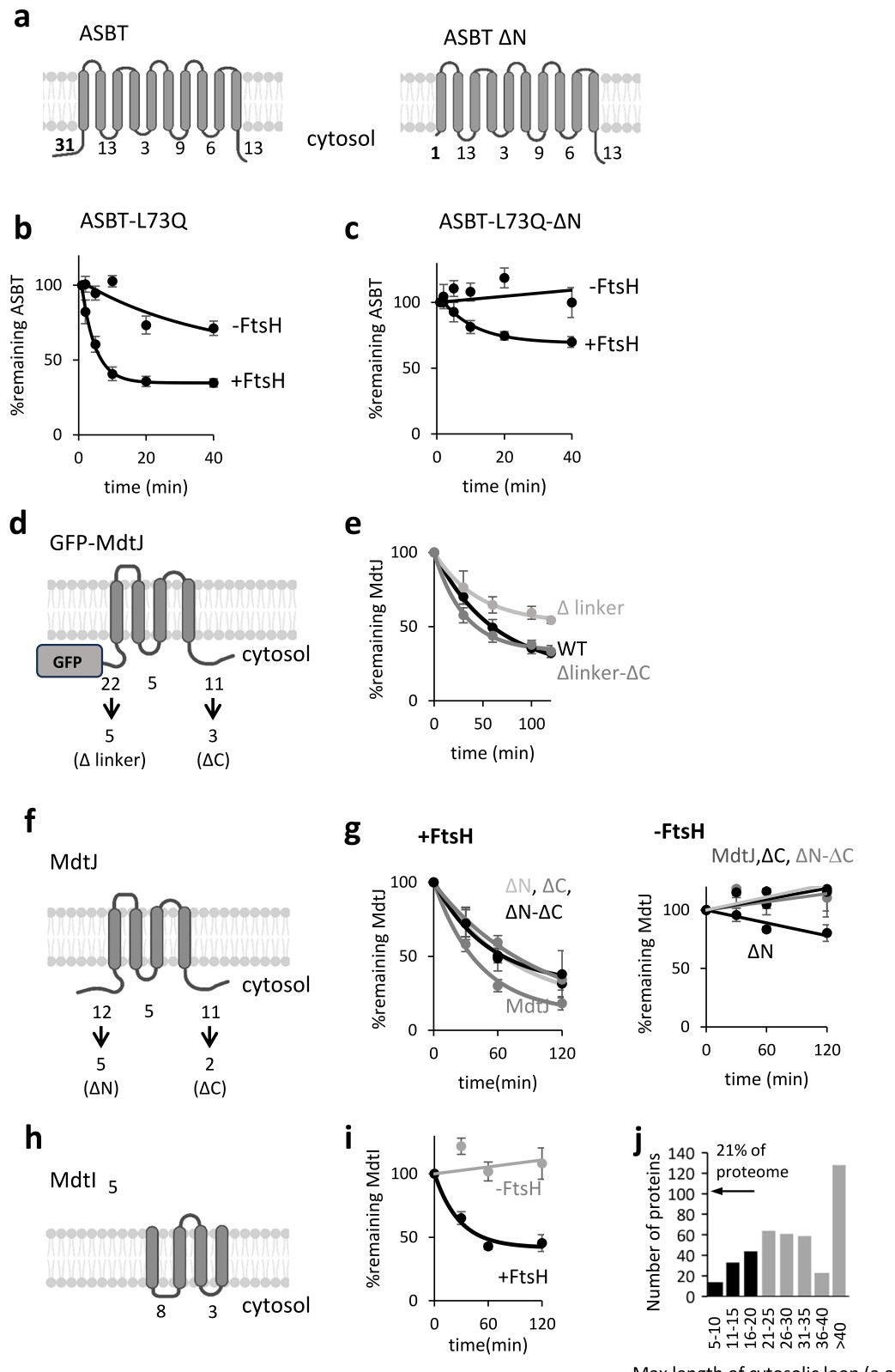

relatively low confidence in most transmembrane parts (pLDDT between 50 and 70), supporting TM dynamics and indicating that the model should be taken with caution (Supplementary Fig. 4e). Nevertheless, the model agrees with the predicted TM boundaries and a kinked part of the centrally-located TM2 that was resolved in a recent structure[43], suggesting that it well captures the hexameric core and topology (Fig. 6a).

Given these uncertainties, we used systematic mutational scanning to identify specific residues involved in MdtJ degradation. We mutated all polar TM residues of FtsH, which we suspected may form polar contacts with substrates, as well as all residues of TM1, which is predicted to face the lipid bilayer (Fig. 6a). Polar residues were replaced with similarly sized apolar ones, and the hydrophobic residues were replaced by ala-nines or leucines (residue numbering was as in Wang et al.[69], adding 3

**Fig. 4 | Effect of cytosolic stretches on the degradation of various proteins by FtsH. a** Topological representation of different ASBT constructs used. Numbers indicate the cytosolic loop lengths according to PDB 4n7x. Created in BioRender. Kalinin, I. (2026) https://BioRender.com/xjnqb4r. **b, c** Degradation of ASBT L73Q (b) and its ΔN mutant (c) in the presence and absence of FtsH. The ASBT L73Q ΔN tail mutant is still degraded by FtsH, although to a lesser extent. Shown are means ± SEM, quantified from four biological repeats. Raw data are presented in Supplementary Fig. 3c. **d** Topological representation of different GFP-MdtJ constructs used. The sequences of wild-type and mutant termini are given in Supplementary Fig. 3d. Numbers indicate cytosolic loop lengths as predicted by the ΔG predictor (Hessa et al. 2007[87]). Created in BioRender. https://BioRender.com/4t4vlhk. **e** A GFP-MdtJ variant devoid of long cytosolic loops gets degraded in an FtsH-dependent manner. Deletion of the loop between GFP and MdtJ slows down degradation, while deleting also the C-terminal tail restores degradation. Shown are means ± SEM, quantified from at least three biological repeats (see Source Data file for exact n). Raw data are shown in Supplementary Fig. 3d. **f** Topological

representations of different MdtJ constructs used. Numbers indicate cytosolic loops length as predicted by the ΔG predictor (Hessa et al. 2007[87]). The sequences of wild-type and mutant termini are given in Supplementary Fig. 3g. Created in BioRender. Kalinin, I. (2026) https://BioRender.com/4t4vlhk. **g** FtsH-dependent degradation of MdtJ without the N and C-terminal tail. Raw data are shown in Supplementary Fig. 3f. **h** Topological representation and loop length of MdtI. MdtI naturally lacks long cytosolic loops. Created in BioRender. Kalinin, I. (2026) https://BioRender.com/4t4vlhk. **(i)** MdtI is degraded in an FtsH-dependent manner. Raw data are shown in Supplementary Fig. 3h. Shown are means ± SEM, quantified from at least three biological repeats (see Source Data file for exact n). **j** A histogram showing the distribution of the length of the longest cytosolic loop in multipass membrane proteins with high confidence topology predictions (n = 426). Approximately 21% of multipass membrane proteins have minimal ( < 20 aa) or absent cytosolic extensions (black). In all panels quantifying degradation kinetics, Solid lines depict nonlinear regression fits. Source data are provided as a Source Data file.

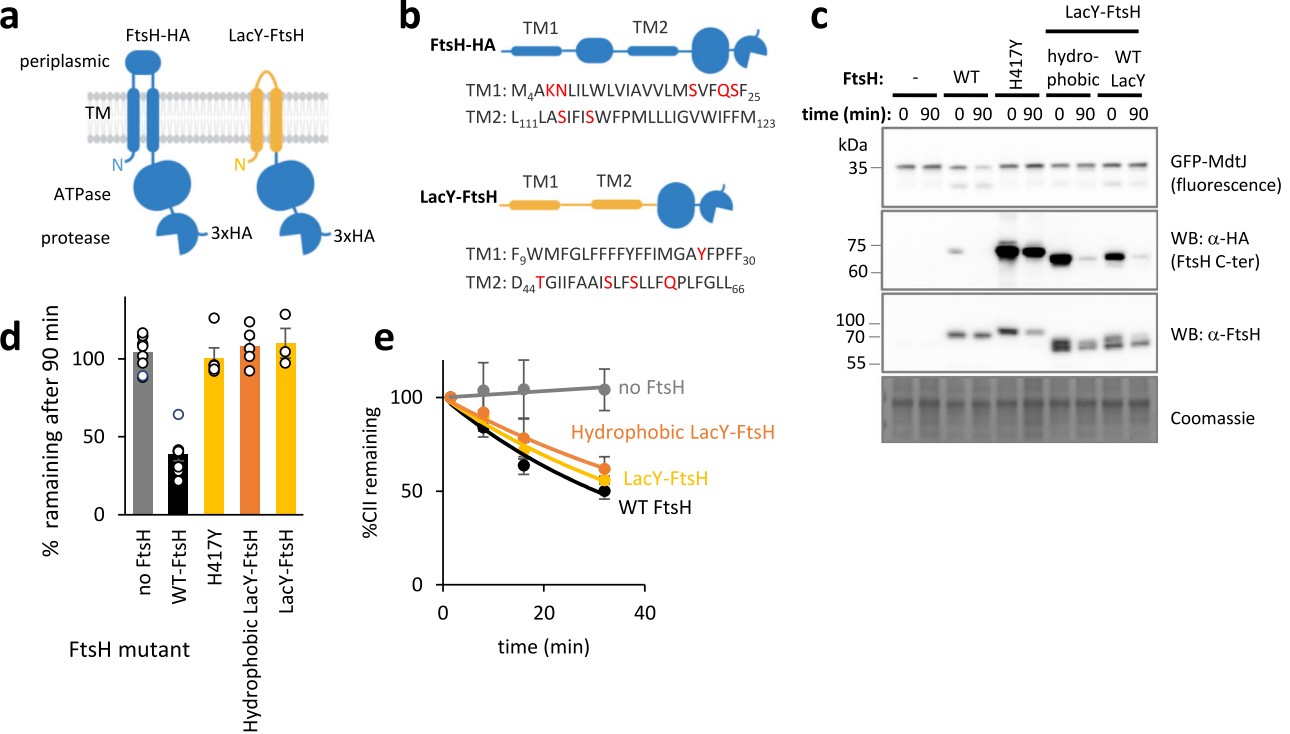

**Fig. 5 | LacY-FtsH chimeras do not degrade MdtJ. a** Topological representation of FtsH and LacY-FtsH, both with a C-terminal HA tag. The membrane was created using BioRender. Kalinin, I. (2026) https://BioRender.com/50mmo62. **b** Sequences of TMs 1 and 2 of FtsH and LacY. The highlighted residues were mutated when indicated. FtsH residue numbering is as in Wang et al. (1998), adding 3 residues to the N-terminus of FtsH compared to the UniProt sequence. The mutations to LacY TMs, generating 'hydrophobic LacY' were Y26V, T45V, S53A, S56A, and Q60A. **c** Activity of FtsH variants in degradation of MdtJ and of the FtsH C-terminal HA tag. The degradation experiment was done by translation shutoff, similar to Fig. 2a, with additional detection of FtsH and its C-terminus by FtsH and HA antibodies, respectively. H417Y is a catalytically inactive FtsH mutant. Coomassie staining is

shown as a loading control. LacY-FtsH chimeras are active proteases, as observed by self-processing of the HA-tag, but cannot degrade MdtJ. **d** Quantification of MdtJ degradation by FtsH mutants from Fig. 5c. Swapping the TMs of FtsH with LacY TMs renders FtsH inactive towards MdtJ. Shown are averages ± SEM from at least three biological replicates. **e** LacY-FtsH chimeras are active proteases, as observed by the degradation of the cytosolic substrate CII. CII degradation was followed by radioactive pulse-chase followed by autoradiography, and quantification of three experiments is shown (means ± SEM) (see Source Data file for exact n). Representative gels are shown in Supplementary Fig. 4c, d. Source data are provided as a Source Data file.

residues to the N-terminus compared to the UniProt sequence). Most mutants were fully active when assessed for protein expression, self-processing activity, and MdtJ degradation. However, four mutations, M4A, A5L, S20A and K6L, inactivated FtsH (Fig. 6d, e). Notably, these mutants displayed reduced FtsH expression and, in some cases, got degraded, suggesting that they perturbed FtsH structure and stability rather than affecting FtsH in a specific manner (Supplementary Fig. 4f, g).

To examine the roles of the polarity of K6 and S20, we further substituted them to isolate mutants with intact expression. When S20

was replaced with valine, asparagine, or threonine, all mutants retained expression and degraded MdtJ, indicating that the polarity of S20 played no role in MdtJ recognition (Supplementary Fig. 4h, i). By contrast, the positive charge of K6 appeared important for folding, as replacing it with a positively charged arginine preserved activity, while a neutral alanine or leucine reduced FtsH expression and abolished activity (Fig. 6d). We suspected that the positive charge of K6 might be important to maintain the correct membrane insertion and orientation of TM1, in accordance with the "positive inside" rule[70].

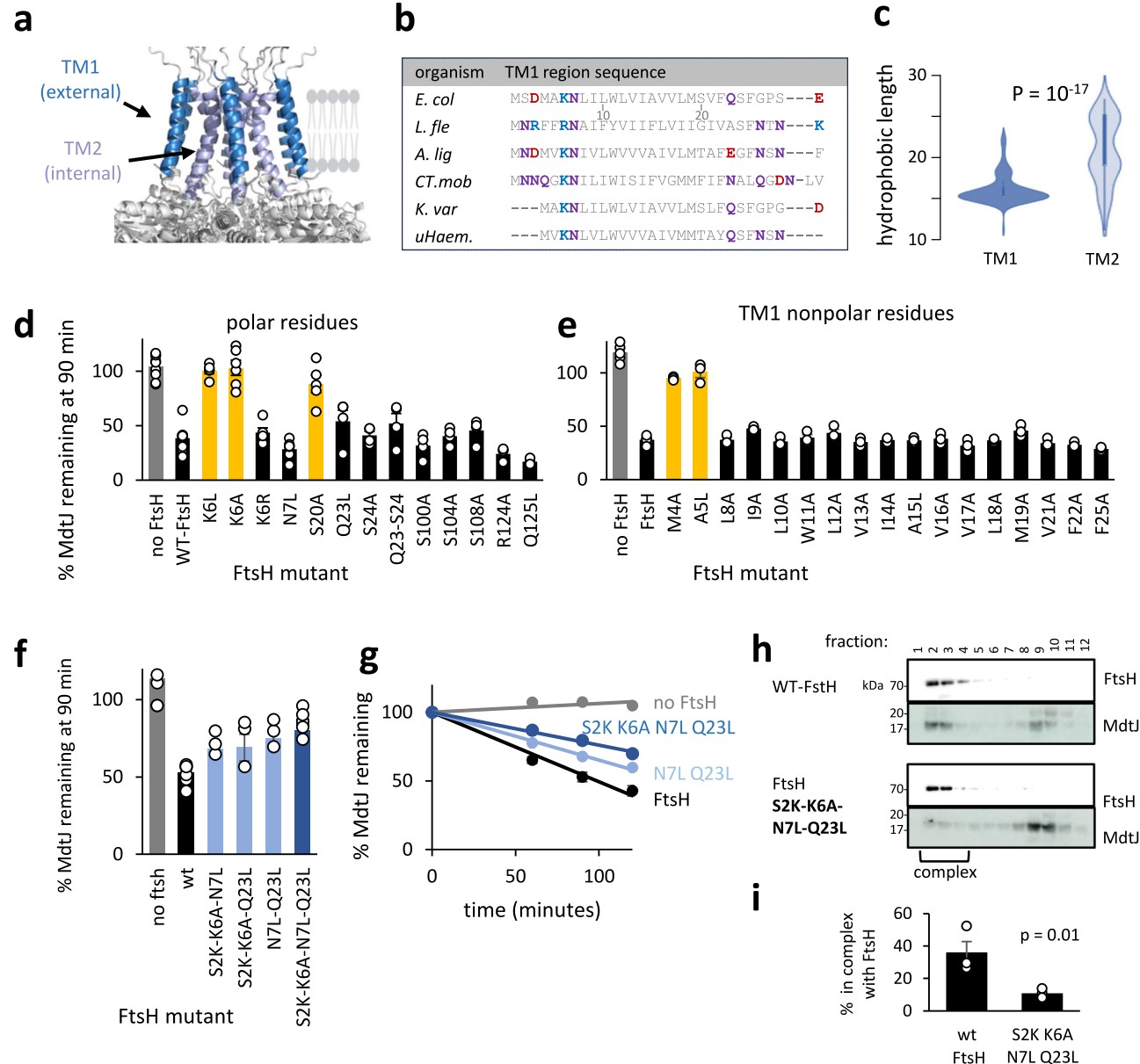

**Fig. 6 | The FtsH transmembrane domain facilitates MP QC. a** Architecture of the hexameric FtsH transmembrane domain in an AlphaFold3 model. TM1 surrounds a central hexameric TM2. **b** Sequences of TM1 in selected bacterial FtsH homologs highlighting the conservation of highly polar residues (colored as in Fig. 1b), which define a short hydrophobic stretch in this TM. Shown are sequences of diverse bacterial species (*E. coli, L. fleischmannii, A. lignieresii, Ca.T. mobilis, K. variicola, uncultured Haemophilus sp.*). For more species, see Supplementary Fig. 6. **c** Distribution of the lengths of the stretch of hydrophobic residues in TM1 and TM2 among 94 FtsH clustered nonredundant homologs. Inner boxplots show IQR (25th–75th percentiles); whiskers extend to the most extreme values within 1.5×IQR. TM1 is typically shorter than TM2, and its hydrophobic length shows less variability. *P*-value was calculated by the two-sided Mann-Whitney test. The multiple sequence alignment of the homologs is given in Supplementary Fig. 6a. **d, e** Quantification of MdtJ degradation by FtsH polar residue mutants (including moderately polar serines) (**d**) and TM1 mutants (**e**). The amount of MdtJ left after 90-min translation shutoff is shown. Most residues are dispensable for MdtJ

degradation, except for K6, S20, M4, and A5. Bars show averages ± SEM of at least three biological replicates (see Source Data file for exact n). Representative gels are shown in Supplementary Fig. 4f, g. **f** Quantification of MdtJ degradation by FtsH mutated combinatorially in positions harboring polar residues near TM1 boundaries. Extending the hydrophobicity of TM1 inhibits MdtJ degradation. Bars show averages ± SEM of at least three biological replicates (see Source Data file for exact n). Representative gels are shown in Supplementary Fig. 5a. Notably, the FtsH gene was switched to anhydrotetracycline-controlled plasmids to equalize the expression of mutants. **g** Degradation kinetics of MdtJ by FtsH TM1 variants. Solid line: the data were fitted to a linear equation. The datapoints represent two independent biological repeats. Raw data are presented in Supplementary Fig. 5b. **h** Size exclusion chromatography shows that the FtsH S2K-K6A-N7L-Q23L mutations perturb the interaction of FtsH with MdtJ, similar to Fig. 3h. **i** Quantification of the percent of MdtJ found in a high molecular weight complex, as determined from densitometry of three repetitions of the experiment presented in (**h**). *P*-value was calculated by a two-sided t-test. Source data are provided as a Source Data file.

Such a role might be facilitated by a positive charge at any position along the cytoplasmic tail flanking the TM. Indeed, a lysine introduced in positions 2, 4, or 5 restored the activity of the K6A mutant (Supplementary Fig.4j, k), indicating that the exact location of the lysine is not important for MdtJ degradation. These substitutions also

indicated that the precise identities of M4 and A5 are not crucial, as they can be replaced with lysines. Taken together, while there might be some role for individual FtsH transmembrane domain residues, their exact location is dispensable for the degradation of MdtJ.

Given the limited effects of individual TM1 mutations, we hypothesized that a more general architectural feature, rather than a specific residue, might contribute to substrate recognition. A close look at TM1 showed that it is flanked by highly polar residues on both ends: K6 and N7 on the cytosolic side, and Q23 on the periplasmic side (Fig. 6b). The polar caps define a relatively short hydrophobic core of only 15 residues. This is shorter than the ~20 residues typically required to span the ~30 Å hydrophobic thickness of the membrane, suggesting that TM1 may experience hydrophobic mismatch with the membrane. This feature is conserved among bacterial FtsH homologs and observed in several eukaryotic homologs, but not in TM2 (Fig. 6b, c, Supplementary Fig. 6). Hydrophobic mismatch can perturb local membrane properties, introducing local polarity, which may thin the bilayer. These features may promote the interaction of FtsH with exposed polar residues in misfolded membrane proteins[71,72], or facilitate their extraction from the membrane[73,74].

To test the role of the short TM, we artificially extended the hydrophobic region of TM1 by engineering double and triple hydrophobic substitutions to K6, N7, and Q23 (along with S2K as a suppressor for K6A). All engineered TMs moderately inhibited MdtJ degradation (Fig. 6f). Further analysis of two mutants, N7L-Q23L and S2K-K6A-N7L-Q23L, showed that they retained self-processing and cytosolic substrate degradation (Supplementary Fig. 5c, d), but displayed a reduced rate of MdtJ degradation (Fig. 6g). Remarkably, size exclusion chromatography indicated that the S2K-K6A-N7L-Q23L mutant, which degraded MdtJ most slowly, lost its ability to form a stable complex with MdtJ (Fig. 6h, i).

Collectively, while no single lipid-facing residue appeared essential, the architecture of TM1, particularly its polarity or hydrophobic length, facilitates the engagement of misfolded MPs by FtsH.

## Discussion

Here, we reveal that lipid-facing polar residues serve as a major degron, a sequence feature that directs degradation by FtsH. This provides an explanation for the specificity toward misfolded MPs, since folded and assembled MPs typically bury membrane-embedded polar residues in their cores, away from the lipid bilayer. Lipid-exposure of polar residues may thus provide a general way to recognize the large diversity of misfolded membrane proteins. An analogous but inverted principle guides the QC of soluble proteins, where the aberrant exposure of core hydrophobic residues attracts QC[75]. Our findings suggest that exposing polar residues to lipids is toxic to the cell, explaining the need for their elimination by QC. The recognition of polar residues is emerging as an important principle for QC along the membranes of the eukaryotic secretory pathway as well. Recent evidence suggests that this principle drives recognition by ER membrane chaperones[30–32,76], an intramembrane protease[22], and ER and Golgi-localized E3 ligases[23–25,33]. Interestingly, our findings suggest that this principle may be shared by more than one protease in *E. coli*, as degradation of ASBT mutants was not fully abolished in the absence of FtsH.

Polar residues are not only essential for recognition by FtsH, but they are sufficient on their own. A single glutamine on the surface of the folded ASBT caused its degradation at rates comparable to or even faster than a genuine misfolded protein MdtJ. This observation showcases the high sensitivity of the QC system to lipid-facing polar residues. Interestingly, the finding that many endogenous polar residues of MdtJ were dispensable for degradation suggests that the position of these residues plays a role in recognition. While the mechanistic explanation for this positional effect is unclear, it is plausible that the different positions display variable levels of lipid exposure in the dynamic misfolded state[77]. Such a reliance on properties beyond polarity might improve the identification of misfolded MPs. Notably, polar residues may also facilitate degradation by accelerating the extraction of MPs from the membrane by FtsH[50].

Recognition by FtsH-family proteases has classically been attributed to unstructured polypeptide stretches located outside the membrane. These stretches, accessible to the ATPase domain (which in *E. coli* is on the cytosolic side), are thought to provide a handle far enough from the membrane to allow the ATPase to initiate pulling the protein out of the bilayer. Our results show that the presence of lipid-facing polar residues can bypass the requirement for these cytosolic stretches. This suggests that these two features, extramembrane stretches and membrane-embedded polarity, define separate routes to recognition by FtsH. This distinction could help explain why the FtsH transmembrane domain is dispensable for some substrates, yet essential for others.

How the FtsH ATPase initiates pulling in the absence of long cytosolic regions remains unclear. One possibility is that parts of the substrate TMs may transiently unfold, forming "cryptic" loops that extend from the membrane, as recently suggested for certain soluble FtsH substrates[65,66]. Another is that the flexible linker between FtsH's ATPase and transmembrane domains[78] enables the ATPase to move closer to the membrane surface. Regardless of the precise mechanism, the ability of QC factors to recognize substrates entirely within the membrane is likely advantageous, as this is where helix-bundle MPs fold and misfold. Indeed, in multiple degradative systems, the transmembrane regions of substrates, E3 ligases, or their adapter proteins play a decisive role in substrate recognition[23–25,79–83], highlighting the importance of membrane-embedded cues across organisms.

How does FtsH detect exposed polar residues within the membrane? Analysis using MdtJ as a substrate suggests that recognition does not depend on a single critical residue in the FtsH transmembrane domain, but rather on the overall architecture of TM1, where several polar residues play partially redundant roles. TM1 is flexible and likely partially exposes multiple faces to the lipid. This raises the possibility that, much like its substrates, FtsH itself presents lipid-facing polar residues. These residues could act indirectly, for example by recruiting an adapter protein, but a more direct role is plausible: the polar "caps" of TM1 create a short hydrophobic span that is mismatched to the thickness of the bilayer. Hydrophobic mismatch can perturb the local lipid organization, causing membrane thinning − a phenomenon recently demonstrated for FtsH reconstituted in lipid bilayers[62]. Such mismatch could aid substrate degradation in several ways: (i) promoting broad interactions between FtsH and other MPs, sequestering the polarities in the interface, shielded from the unfavorable lipid exposure[33,71,72]; (ii) facilitating extraction of the protein from the membrane by lowering the energetic barrier for protein retrotranslocation[73,74]; and (iii) potentially positioning FtsH in thinner membrane microdomains where misfolded MPs are enriched. These mechanisms are not mutually exclusive, and additional factors, including possible adaptors, may further sharpen substrate specificity. The conservation of short hydrophobic TM1s across many FtsH homologs, together with our finding that altering polar residues in either FtsH or MdtJ disrupts their interaction, supports a model in which membrane polarity directly contributes to substrate engagement.

To conclude, our results identify lipid-facing polar residues as potent drivers of MP QC, providing a unifying principle that explains how FtsH and potentially other QC factors can selectively target misfolded MPs. Defining how this integrates with other degradative pathways, and how it shapes the cellular response under proteotoxic stress, will be important directions for future research.

## Methods
### Bacterial strains
*E. coli* DH5α was used for plasmid propagation. BL21 (DE3) and the Δ*ftsH* strain AR5090 (DE3) Δ*ftsH3::kan, sfhC21/F'lacIq*[60] were used for

protein expression and for radioactive pulse-chase degradation assays. Toxicity assays of the MdtJ protein were done using the *E. coli* Δ*ftsH* strain AR5090 (DE3)[60].

## Plasmids and cloning

The list of plasmids used is provided in Supplementary Table 1. All mutants were engineered by PCR-based techniques such as Quick-Change, blunt-end ligation, and Gibson assembly, and their sequences were verified. pZ ASBT, a pET derivative expressing Cys-less ASBT, was described in Zhou et al. 2014[57], it was further engineered to replace the kanamycin resistance gene with ampicillin resistance. pBAD/HisB-sfGFP (TIR^STD) was a kind gift from Daniel O. Daley[84]. The pET19b (Novagen) expressing MdtJ and MdtI were described in ref. 60. Both MdtJ and ASBT were Cys-less versions.

The *MdtJ* and *MdtI* genes were amplified from the pET 19b MdtJI plasmids. Post-amplification, they were cloned into the pBAD/HisB-sfGFP (TIR^STD) vector using Gibson Assembly method. The primers used for the cloning process were as follows:

MdtJ Forward Primer:GATTACACATGGCATGGATGAACTCTA-CAAAATGCAACCCCGTCAATATACTTCAC. MdtJ Reverse Primer: CA AAACAGCCAAGCTTCGAATTCTTATTAGTTATTTGATGAAATATTTA CTG. pBAD forward primer: TAAGAATTCGAAGCTTGGCTGTTTTGG. pBAD reverse Primer: TTTGTAGAGTTCATCCATGCCATGTG. Similarly, the *MdtI* gene was extracted from the pET19b MdtJ+MdtI.

Plasmid pSTD240 cII was a kind gift from Yoshinori Akiyama. The cII gene was cloned into the pZ plasmid. cII forward primer: ATGG TTCGTGCAAACAAACGCAACG. cII reverse primer: TCAGAACTCCA TCTGGATTTGTTC. pZ forward primer: CTGAACAAATCCA-GATGGAGTTCTGATACTCGAGCACCACCACCACCACC. pZ reverse primer: CGTTGCGTTTGTTTGCACGAACCATGGTATATCTCCTTCTTA AAGTTAAAC

The *ftsH* gene was amplified from the MG1655 *E. coli* strain genome. The amplification included a 200 bp region upstream of the gene, serving as the native promoter, and a 110 bp region downstream, acting as the natural terminator. The following primers were utilized for the amplification: Forward primer: AAATTCGCTCCCTGTTTAC-GAAGGTC. Reverse primer: ACCCTGGGCAAAGAGTTTCATGATG. After amplification, the gene was cloned into the pACYC184 plasmid using the Gibson Assembly method[85]. Additionally, a triple-hemagglutinin (3xHA) tag was appended to the C-terminus of the protein, facilitating subsequent detection.

## ASBT purification and thermal stability assay

ASBT purification was done similarly to Zhou et al.[57] with slight changes. Cellular growth and harvesting: The indicated plasmids were transformed into the FtsH deletion strain AR5090 (DE3). A single colony was grown in LB medium supplemented with 0.5% Glucose overnight and then diluted 100-fold into 1000 mL of the same medium. Cells were grown at 37 °C to an optical density of ~0.6. Overexpression of the protein was induced by 0.5 mM of isopropyl β-D-1-thiogalactopyranoside (IPTG) and the culture was grown at 20 °C overnight. An equal volume of cells corresponding to 500 OD units was collected and washed with 100 mL of wash buffer (20 mM HEPES, pH 7.5, 150 mM NaCl). Subsequently, the cells were pelleted and resuspended in 50 mL of lysis buffer (20 mM HEPES, pH 7.5, 150 mM NaCl, 10%(v/v) glycerol, 2 mM β-mercaptoethanol)

**Membrane preparation.** Cells were thawed in a water bath, Turbo DNase (Jena biosciences) and 1 mM phenylmethanesulfonyl fluoride (PMSF) were added and cells were lysed using an EmulsiFlex homogenizer (three passes). The lysed cells were centrifuged to discard cell debris. The supernatant was collected and centrifuged to collect membranes (100,000 × *g*, 60 min at 4 °C). The pellet containing membranes was resuspended using homogenizer in 8 mL lysis buffer. Aliquots of 2 mL were snap frozen.

**ASBT purification.** The supernatant was loaded onto a talon metal affinity column, washed twice with 3 mL wash buffer (20 mM HEPES, pH 7.5, 150 mM NaCl, 10%(v/v) glycerol, 2 mM β-mercaptoethanol, 20 mM imidazole 0.2% DDM) and eluted with elution buffer (20 mM HEPES, pH 7.5, 150 mM NaCl, 10%(v/v) glycerol, 2 mM β-mercaptoethanol, 300 mM imidazole, 0.2% DDM). Protein concentration was measured by nanodrop and calculated by dividing by the mass extinction coefficient (1.19 mL mg$^{-1}$ cm$^{-1}$ for WT and L73Q mutant, 0.943 mL mg$^{-1}$ cm$^{-1}$ for ΔTM10). Typical protein yields were 0.2-0.4 mg/mL. Equal amounts of protein were dialyzed twice for at least 1 h each time in lysis buffer. Thermal stability was assayed for 2 replicates in a volume of 10 μL each. Changes in the intrinsic fluorescence were measured by nanoDSF (Prometheus NT.48). Lysis buffer supplemented with 0.2% DDM served as a blank.

## Crosslinking by dibromobimane (dBBr)

**Cell growth and harvesting.** A total of 500 mL of cells were grown, harvested and washed as described in the ASBT purification section and resuspended in 50 mL lysozyme buffer (150 mM NaCl, 30 mM Tris−HCl, pH 8.0, 10 mM EDTA, 1 mg/mL lysozyme, 0.5 mM phenylmethanesulfonyl fluoride (PMSF)). The suspension was snap frozen.

**Crude membrane preparation.** The frozen cells were thawed in a room-temperature water bath and then moved to 37 °C shaker with intermittent shaking for 10 min. 180 mL of DNase solution (15 mM MgSO$_4$, Turbo DNase, 0.5 mM PMSF) was added, and the mixture was further agitated for 15 min. After incubation, the cell suspension was immediately transferred to an ice bath for 10 min. Crude membranes were isolated by centrifugation at 20,000 × *g* for 30 min at 4 °C. The resulting pellet was resuspended in 3.8 mL of ASBT buffer (20 mM HEPES, pH 7.5, 150 mM NaCl, 10% v/v glycerol, 2 mM β-mercaptoethanol) using a homogenizer. The membrane suspension was aliquoted into six separate tubes and rapidly frozen in liquid nitrogen for subsequent analyses.

**Membrane washing and crosslinking.** One tube for each mutant was thawed in a water bath. The membranes underwent two wash cycles using 1 mL of membrane wash buffer (20 mM HEPES pH 7.5, 150 mM NaCl) to remove any residual reducing agents. Subsequently, membranes were resuspended in 1.1 mL of the same buffer. Aliquots of 500 μL were divided into control (-dBBr) and experimental (+ dBBr) tubes. All subsequent steps were conducted in a dark environment to prevent photobleaching. Each tube received 5 μL of freshly prepared 0.2 M dibromobimane (dBBr, Sigma) or dimethyl sulfoxide (DMSO) and incubated for 1 h at 4 °C with gentle agitation. The crosslinking reaction was quenched with 50 μL of 0.2 M N-ethylmaleimide (NEM) for 30 min.

**ASBT purification.** Membrane proteins were solubilized in 0.5 mL membrane solubilization buffer (20 mM HEPES, pH 7.5, 150 mM NaCl, 10% glycerol, 2% DDM, 20 mM imidazole, 1 mM PMSF) through agitation (1000 RPM, 20 °C) for 5 min, followed by a 15 min incubation on ice. The mixture was then centrifuged at 20,000 × *g* for 5 min at 4 °C. The supernatant (800 μL) was incubated with 30 μL of Talon beads (Takara Bio) at 4 °C for 1 h with continuous shaking. The beads were washed twice with wash buffer (20 mM HEPES pH 7.5, 150 mM NaCl, 10% glycerol, 0.2% DDM, 20 mM imidazole) and eluted with 60 μL of elution buffer (1.5x non-reducing sample buffer, supplemented with 450 mM imidazole).

The samples were subjected to SDS-PAGE gel electrophoresis and initially imaged using a gel documentation system with a 365 nm excitation wavelength (BioRad). Post-imaging, the gel was stained with Coomassie Instant Blue and dried. If needed, quantitative analysis was performed to ensure equal protein loading for a second round of gel electrophoresis.

## ASBT degradation by radioactive pulse-chase

A smear of colonies from a fresh transformation of ΔftsH AR5090(DE3) or BL21(DE3) harboring the indicated ASBT expressing plasmids was grown overnight in M9 medium supplemented with 1 g/L CSM-Met (MP biomedical, cat 4510712), 100 μg/mL thiamine, 0.4% glycerol, 2 mM MgSO$_4$, 0.1 mM CaCl$_2$, 0.5% glucose and 100 μg/mL ampicillin. The culture was then back-diluted to 0.1 OD into the same medium without glucose and grown at 37 °C to mid-log phase (OD$_{600}$ of ~0.5). Cultures were induced with 0.1 mM IPTG for 10 min, followed by 15 min of incubation with 0.2 mg/mL rifampicin to halt transcription other than T7 polymerase-dependent transcription. Proteins were labeled with 15 μCi [$^{35}$S]Met (PerkinElmer) for 1 min, then mixed with a high excess (2 mM) of non-radioactive methionine. Samples of 0.3 mL were taken at variable chase times and mixed with 0.75 mL ice-cold PBS to rapidly cool them. Cells were harvested (5 min, 10,000 × $g$ at 4 °C) and resuspended in 100 μL sample buffer. Protein separation was conducted using 12% polyacrylamide gel electrophoresis. Gels were subsequently dried, visualized by autoradiography, and quantified.

## MdtJ degradation by radioactive pulse-chase

The experiment was done as described for ASBT with ΔftsH AR5090(DE3) strain harboring the indicated pET and pACYC plasmids. Samples of 0.5 mL were taken at variable chase times and mixed with 0.5 mL ice-cold 20% TCA to stop the reaction. TCA precipitation was performed: samples were kept on ice for at least 30 min and then harvested (20 min, 20,000 × $g$ at 4 °C). Samples were washed by 0.8 mL of ice-cold acetone and harvested (10 min, 20,000 × $g$ at 4 °C) and resuspended in 100 μL sample buffer supplemented with 100 mM of Tris-HCl pH 6.8. Protein separation was conducted using 12% polyacrylamide gel electrophoresis. Gels were subsequently dried, visualized by autoradiography, and quantified.

## CII degradation analysis

The procedure was analogous to the one described for the ASBT mutants. However, the E. coli ΔftsH AR5090 (DE3) strain was co-transformed with the pZ-cII plasmid and the specified pACYC FtsH plasmid. The growth medium was supplemented with 0.5% glucose to reduce CII expression. When FtsH was under tetracycline promoter, 50 ng/ml anhydro tetracycline was added to the medium.

## GFP-MdtJ degradation assay

A smear of colonies from a fresh transformation of the ΔftsH AR5090 (DE3) harboring the indicated FtsH and GFP-MdtJ expressing plasmids, was grown overnight in LB medium supplemented with 50 μg/mL ampicillin, 17 μg/mL chloramphenicol and 0.5% Glucose. The cultures were then back-diluted to 0.1 OD into the same medium without glucose and were grown at 37 °C to mid-log phase (OD$_{600}$ of ~0.5). GFP-MdtJ expression was induced by adding 0.1 % arabinose for 30 min. When FtsH was under tetracycline promoter, 50 ng/ml anhydro tetracycline was added. Cultures were further treated with 200 μg/mL spectinomycin to halt translation, and samples (0.5 mL) were collected and immediately cooled on ice. Additional sets of samples were collected at the indicated times. Cells were harvested (5 min, 10,000 × $g$ at 4 °C), washed with 0.6 mL PBS, resuspended with 150 μL lysozyme buffer (150 mM NaCl, 30 mM Tris–HCl pH 8, 10 mM EDTA, 1 mM DTT, 1 mg/mL lysozyme (sigma) and 1x cOmplete Protease Inhibitor (roche)) and frozen at −20 °C. Cells were disrupted by thawing at 25 °C for 5 min, followed by shaking at 37 °C for 10 min. Then, 0.9 mL of DNase solution (15mM MgSO$_4$, 1mM DTT, and 1 mM PMSF, 1 x Turbo DNase (Jena biosciences)) was added, and the samples were allowed to shake at 37 °C for 10 min before transferring to ice. Crude membranes were collected by centrifugation at 4 °C, 20,000 × $g$ for 20 min and the pellet was resuspended in 50 μL of GFP sample buffer (50 mM Tris 8.0, 2% SDS, 10% glycerol, 20 mM DTT, bromophenol blue).

Alternatively, after cell harvest, cells were directly resuspended with 100 μL of GFP sample buffer. Both methods gave the same results.

Equal amounts of cells were loaded on a 12% polyacrylamide gel. In gel fluorescence was visualized by Typhoon and quantified, followed by Coomassie staining. FtsH visualization was done by western blotting with anti-HA antibody (Sigma-Aldrich, H9658) or anti-FtsH antibodies (the National BioResource Project (NBRP)-E. coli, Japan) diluted 1:10,000.

## MdtJ toxicity

A smear of colonies from a fresh transformation of ΔftsH AR5090 (DE3) harboring an FtsH plasmid and the indicated MdtJ expressing plasmids was grown overnight in LB medium supplemented with 100 μg/mL ampicillin and 0.5% glucose. The culture was then back-diluted to 0.1 OD into the same medium without glucose and grown at 37 °C to mid-log phase (OD$_{600}$ of ~0.5). Five tenfold serial dilutions were prepared from the mid-log cultures, with the highest density having an OD$_{600}$ of 0.1. The serial dilution was spotted (3 μL) on LB agar–ampicillin plates containing indicated amounts of IPTG and were grown overnight at 37 °C.

## FtsH - MdtJ interaction assay

Cell growth was done separately for MdtJ and FtsH-expressing cells. For FtsH, pET21a encoding FtsH-His6 was transformed into the FtsH deletion strain AR5090 (DE3). A single colony was grown in Luria-Bertani (LB) medium supplemented with 0.5% Glucose overnight and then diluted 100-fold to 500 (WT) or 1000 (mutant) mL of the same medium. Cells were grown to an optical density at 600 nm of ~1 at 37 °C. Overexpression of the protein was induced by the addition of isopropyl β-D-1-thiogalactopyranoside (IPTG) to a final concentration of 0.15 mM at 37 °C for 4 h and then left at 4 °C overnight. A volume of cells corresponding to 500 OD units was collected and subjected to a washing step with 100 mL of wash buffer (20 mM HEPES, pH 7.5, 150 mM NaCl). Subsequently, the cells were pelleted (10 min at 5000 g) and resuspended in 30 mL of lysis buffer (20 mM Tris, pH 8, 400 mM NaCl, 1 mM EDTA, 2 mM DTT, 10% glycerol). For MdtJ, the procedure was similar with minor modifications. pET19b MdtJ-His6 was used. The culture volume was 2 L, and induction was done by 0.3 mM IPTG for 5 h.

**Membrane preparation.** Cells were thawed in a water bath, added with Turbo DNase and 0.5 mM PMSF, and lysed using Emulsiflex cell disruptor 2 times. The lysed cells were centrifuged (10,000 × $g$, 15 min) to discard cell debris. The supernatant was collected and centrifuged to collect membranes (150,000 × $g$, 60 min at 4 °C). The membrane pellet was resuspended using homogenizer in 5/10 mL buffer (for FtsH:10 mM Tris pH = 8, 200 mM NaCl, 2 mM MgCl$_2$, 0.5 mM ATP, 20% glycerol, 5 mM DTT; For MdtJ: 10 mM Tris 8, 2 mM DTT, 2 mM Mg, 20% glycerol, 300 mM NaCl), and frozen with liquid nitrogen, with each mL of membranes corresponding to 150–200 optical density units.

**Interaction.** Membranes were thawed quickly and kept on ice. 25 μL of FtsH membranes were supplemented with 2.5 μL of activity buffer (10 mM ATP, 20 mM MgCl$_2$, 0.05 mM Zn acetate, and 0.05 mM Zn chloride) and incubated for 5 min at 4 °C. Next, 7 μL of MdtJ membranes and 35 μL of 10 mM Tris-HCl pH 8.0 were added. The mixture was subjected to three freeze–thaw cycles using liquid nitrogen to promote membrane vesicle fusion and interaction of embedded proteins and then solubilized in 515 μL of gel filtration buffer (20 mM Tris-HCl pH 8.0, 150 mM NaCl, 1 mM DTT, 1 mM MgCl$_2$, 0.025% NP-40; all buffers were filtered before use) supplemented with 10 μL of 10% NP-40 for 10 min on ice. Samples were clarified by centrifugation at 20,000 × $g$ for 20 min at 4 °C to remove insoluble material. The supernatant was loaded onto the gel filtration column, an ÄKTA system equipped with a column (Superose 6 increase 10/300 GL (Cytiva)),

equilibrated with gel filtration buffer (24 mL column volume). Fractions of 1 mL were collected, and those corresponding to ~2000–170 kDa were analyzed by SDS-PAGE. Western blotting was performed using anti-His antibodies (Invitrogen, MA1-135-HRP) to detect FtsH and MdtJ.

## Software

Gels were quantified by the *ImageQuant^TM* software (Cytiva), following the developer's instructions. Protein structures were visualized using The *PyMOL Molecular Graphics System, Version 2.0 Schrödinger, LLC*. Hydrophobic surface coloring was using a script generated by Hagemans et al.[86] Illustrations were created with *BioRender.com*. TM hydrophobicity was calculated using the $\Delta G$ predictor[87]

## Cytosolic loop length evaluation

Topology predictions for multipass membrane proteins were obtained from Kalinin et al.[88] supplementary data 1, and analyzed using custom Python scripts. Proteins with high-quality orientation predictions, medium to high-quality TM number predictions and certainty in N and C terminal localization were included in the study, totaling 426 proteins. For each protein, the longest cytosolic loop was evaluated.

## TM hydrophobic length analysis

A total of 1000 homologous sequences were collected using *HMMER*[89] against the *nr70_bac* database with the full-length FtsH protein as the query. A subsequent iteration was performed for the N-terminal region of FtsH (residues 1–150). Almost all identified homologs were annotated as ATP-dependent metalloproteases of the FtsH family. The homologs were analyzed using *MMseqs2*[90] to generate a reduced set of sequences, which included 94 sequences (identity ≥60%, coverage ≥80%). Sequence alignment was performed using *MAFFT*[91] and visualized using *Jalview (v2.11)*. For each protein, the length of the hydrophobic stretch within TM1 and TM2 was calculated based on identifying the longest stretch of residues within the aligned TM region that does not contain highly polar residues (defined as K/R/N/Q/H/D/E).

## Reporting summary

Further information on research design is available in the Nature Portfolio Reporting Summary linked to this article.

## Data availability

All data supporting the findings of this study are available within the paper and its Supplementary Information. Source data are provided with this paper.

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

## Acknowledgements
We thank Daniel O Daley, University of Stockholm, Yoshinori Akiyama, Kyoto University and Ming Zhou, Baylor College of Medicine, for kindly providing plasmids. We thank lab members for fruitful discussions, Yael Fridmann Sirkis for help with thermal unfolding, and Maja Rennig and Morten HH Nørholm for help with characterization of ASBT degradation. We are grateful to Gunnar von Heijne for initial financial support (grants by the Knut and Alice Wallenberg Foundation (2017.0323) and the Swedish Research Council (621-2014-3713) to GvH) and discussion, and for critically reading the manuscript. This research was generously supported by research grants from the ISRAEL SCIENCE FOUNDATION (grants no. 2207/21 and 2208/21), the Minerva Foundation with funding from the Federal German Ministry for Education and Research (grant 714813), and the Center for New Scientists at the Weizmann Institute of Science. Illustrations were created with BioRender.com.

## Author contributions
M.C.D., N.R.M., V.V., H.P-Z., and N.F. designed the study and the experiments. M.C.D., N.R.M., V.V., T.O., M.P., and A.B. performed the experiments. M.C.D., N.F., N.R.M., and V.V. conducted the majority of data analysis. A.B.D.B. helped with computational analysis. N.F. and M.C.D. wrote the manuscript. All authors discussed the results and contributed to editing the manuscript. N.F. conceived the study and supervised the work.

## Competing interests
The authors declare no competing interests.
