## [Transparent Peer Review file · Nature Communications]

Membrane-embedded polar residues target membrane proteins for degradation by the quality control protease FtsH

Corresponding Author: Dr Nir Fluman

Version 0:

Reviewer comments:

Reviewer #2

(Remarks to the Author)

I thank authors for seriously and rigorously addressing my major concerns. I am glad to see that the overall quality of the manuscript has been improved after the revision, while a few concerns remain unresolved.

Here, I summarized the strengths and weakness of the revised manuscript:

Major findings of this work:

1. Exposed polar residues on membrane proteins increases their susceptibility to degradation in bacteria (Figs. 2 and 3).
2. Exposed polar residues on membrane proteins increases cellular toxicity (Fig. 3k-this is a very good experiment).
3. FtsH-mediated membrane protein degradation does not necessarily involve the recognition by terminal tails or internal loops (Fig. 4. Disrupted native structure (i.e., unfolded states) of membrane proteins is an enough condition for their degradation.

Remaining concerns:

1. Still, it is not fully convincing that the exposed polar residues are "recognition" elements by FtsH (this is one of the major conclusions in this manuscript). The premise of this conclusion is that the unfolded (or denatured) state of a membrane protein looks like Figs. 1b and 3c. That is, upon denaturation, TM helices remain inserted into the membrane while tertiary contacts are lost. In this scenario, polar residues would be exposed to the hydrocarbon core of the membrane and can possibly enhance recognition by FtsH. However, recent studies suggest that the denatured states can be highly dynamic, involving significant interhelical contacts, exist of TM helices to the water-membrane interfaces, and unfolding of TM helices at the interfaces (Seurig et al., 2019, Nat Chem Biol; van Lehn et al., 2015, eLife; Gaffney et al., 2022 PNAS; Lin et al. 2023 eLife). The latter two phenomena are known to depend on the hydrophobicity of the TM helix. As shown in Figs. S1a and S2c, many TM helices are marginally hydrophobic (even slightly hydrophilic), and the hydrophilicity increases upon nonpolar to polar mutation. So, still the possibility cannot be excluded that the recognition happens at the membrane surface, not at the hydrocarbon core of the membrane. Thus, it can also be plausible that the enhanced degradation/recognition by polar residues stems from the enhanced partition and unfolding of TM helices at the water-membrane interfaces. This aspect is not critically addressed in this work.
2. Related to the point above: Generally, in a given catalytic reaction, the apparent outcome (here, degradation) could be decomposed into K_m (related to the binding of a substrate) and k_{cat} (the barrier towards ATP hydrolysis, active denaturation of substrates, and membrane extraction). I have an impression that in this work, the interpretation of the polar residue effect on degradation seems to be biased towards "binding", ignoring their possible effects (polar/nonpolar mutations on both substrates and FtsH) on the barriers in the major catalytic events (ATP hydrolysis, active denaturation, and membrane extraction).
3. The effect of polar/nonpolar mutations on FtsH is largely not conclusive. Many of these mutations retain activity for

degrading water-soluble proteins. However, the activity for membrane proteins displays complex behaviors, in which the sites of effective changes are located at the residues outside the hydrocarbon core. This result renders it hard concluding that the region where the substrate recognition occurs is the membrane core. Authors acknowledges this point. Exposed polar residues on the substrates require their binding partners on FtsH. But, the lack of unequivocal recognition sites within TM1 in the membrane weakens authors' conclusion.

Minor concerns:

1. Line 81: it is unclear why ASBT is an atypical membrane protein.
2. Fig. 5c: Although H147Y is a catalytically inactive mutant (probably, ATPase activity), this mutant seems to show the terminal cleavage activity (the reduced band intensity after 90 min). Some explanation seems to be needed.
3. Fig. 5c and lines 253-255: LacY TM1/2-FtsH chimera does not degrade MdtJ. However, Akiyama/Ito's work (2000, EMBO J; 2001, Biochemistry) has shown that the same chimera can degrade SecY. So, there should be a caution in generalizing the result.

Reviewer #3

(Remarks to the Author)

The authors have sufficiently addressed all my previous comments, and I applaud them on a much-improved manuscript. The additional data enhance the mechanistic insights and significance of their findings. All experiments are elegantly designed and well-controlled. I therefore recommend speedy publication of this study. I only have a few minor comments:

Minor comments:

In Fig. 2b the data points seem too big for the size of the panel.

Panels in Fig. 2e/2f/3e/S3b and many more: Why are the connecting lines between data points squiggly?

Delete 'where' in: 'These results confirm that the tertiary structure of ASBT-L73Q, especially related to TM3,4, and 9, where is maintained in the native membrane.'

Fig. S2d: The legend should say color coding as in Fig. 3c

Fig 3h+3i: Could the authors please provide a reference for their MdtJ-FtsH interaction assay based on the membrane mixing approach? Is this a standard assay in the E. coli field? Could the authors explain in more detail in the methods section how this works? Do the freeze-thaw cycles lead to membrane fusion, thus allowing the proteins to interact before detergent solubilization?

Fig. 4g needs color coding of the different MdtJ variants

Fig. 6b introducing residue numbering for E. coli FtsH TM1 would help with navigating the mutational data in 6d+e

Reviewer #4

(Remarks to the Author)

The revised manuscript by Chai-Danino et al. extends our understanding of how misfolded/mis-assembled membrane proteins are recognized in the lipid bilayer, with a particular focus on the often understudied bacterial system. As such, it will be of interest to the field and in its revised form seems appropriate for publication. The authors may want to mention a recent study in line with their ideas (<https://www.biorxiv.org/content/10.1101/2025.10.31.685944v1>) in the final revision of their work.

Version 1:

Reviewer comments:

Reviewer #2

(Remarks to the Author)

First, I sincerely thank authors for rigorously addressing my concerns and incorporate their responses into the revision. Authors' arguments as well as the basis of the arguments are convincing. Especially, Fig. 2 (ASBT WT vs L73Q) and Fig. 6h (gel filtration chromatography of MdtJ and its mutants) are wonderful data.

I am still very curious about why the well folded, stable ASBT L73Q is highly prone to degradation by FtsH even if the L73Q mutant is recognized better by FtsH than WT. The architectural or membrane thinning model by FtsH TM1 is very attractive, but I am not sure how the enhanced binding to FtsH can be enough to facilitate the engagement into FtsH's AAA+ hexamer and subsequent membrane extraction and degradation. I understand that further discussions can only be made at a

speculative level as we still know too little about the bilayer effects on membrane protein folds, their dynamics, and conformational features of non-native states of membrane proteins. So, authors do not have to address my curiosity.

In summary, this is a beautiful piece of work and authors did their best within their scope. I have no further concern. I enthusiastically recommend an acceptance of the manuscript without any reservation. This work is a tour de force towards the understanding of membrane protein quality control. Readers including myself have learned and will learn a lot from this work. Thank you to the authors!

Reviewer #2 (Remarks to the Author):

I thank authors for seriously and rigorously addressing my major concerns. I am glad to see that the overall quality of the manuscript has been improved after the revision, while a few concerns remain unresolved.

- We thank the reviewer for the positive assessment. Below, we address each remaining concern in detail.

Here, I summarized the strengths and weakness of the revised manuscript:

Major findings of this work:

1. Exposed polar residues on membrane proteins increases their susceptibility to degradation in bacteria (Figs. 2 and 3).
2. Exposed polar residues on membrane proteins increases cellular toxicity (Fig. 3k-this is a very good experiment).
3. FtsH-mediated membrane protein degradation does not necessarily involve the recognition by terminal tails or internal loops (Fig. 4. Disrupted native structure (i.e., unfolded states) of membrane proteins is an enough condition for their degradation.

Remaining concerns:

1. Still, it is not fully convincing that the exposed polar residues are "recognition" elements by FtsH (this is one of the major conclusions in this manuscript). The premise of this conclusion is that the unfolded (or denatured) state of a membrane protein looks like Figs. 1b and 3c. That is, upon denaturation, TM helices remain inserted into the membrane while tertiary contacts are lost. In this scenario, polar residues would be exposed to the hydrocarbon core of the membrane and can possibly enhance recognition by FtsH. However, recent studies suggest that the denatured states can be highly dynamic, involving significant interhelical contacts, exist of TM helices to the water-membrane interfaces, and unfolding of TM helices at the interfaces (Seurig et al., 2019, Nat Chem Biol; van Lehn et al., 2015, eLife; Gaffney et al., 2022 PNAS; Lin et al. 2023 eLife). The latter two phenomena are known to depend on the hydrophobicity of the TM helix. As shown in Figs. S1a and S2c, many TM helices are marginally hydrophobic (even slightly hydrophilic), and the hydrophilicity increases upon nonpolar to polar mutation. So, still the possibility cannot be excluded that the

recognition happens at the membrane surface, not at the hydrocarbon core of the membrane. Thus, it can also be plausible that the enhanced degradation/recognition by polar residues stems from the enhanced partition and unfolding of TM helices at the water-membrane interfaces. This aspect is not critically addressed in this work.

-We appreciate the reviewer's thoughtful discussion of membrane-protein unfolded states. We agree that misfolded membrane proteins can adopt dynamic topologies. Experimentally defining these states remains extremely challenging and typically forms the basis of a dedicated, multi-year projects (DOI: 10.1038/s41589-019-0356-9; doi: 10.1073/pnas.2109169119; DOI: 10.1073/PNAS.2103674118; doi: 10.1126/science.aaw8208.). Even in these studies, many conclusions rely on simulations and indirect evidence. Experimentally resolving dynamic topologies in vivo is still feasible only for isolated cases. Extending such analyses to three unrelated substrates (MdtJ, MdtI and ASBT) would require methods that do not currently exist at scale. This inherent limitation of the field, rather than a conceptual omission, is why such analyses fall outside the scope of the present study.

While such topological dynamics may occur in unfolded proteins, these considerations do not contradict our central conclusion. Three points support this:

1. Our data provide strong experimental evidence that lipid-facing polar residues can act from within the membrane.
2. We have explicitly acknowledged the likely heterogeneity of unfolded states and already discussed these dynamics throughout the manuscript.
3. The reviewer's concern about TM hydrophilicity may overestimate the extent to which these helices behave as marginally hydrophobic in vivo, and experimental evidence suggests that MdtJ helices remain robustly inserted.

Below, we provide detailed responses to each of these three points.

1. Our experimental data strongly support a contribution from polar residues acting within the membrane.

(a) ASBT-L73Q behaves as a folded substrate with a fixed transmembrane structure. ASBT-L73Q retains its native fold and stability, as shown by thermal denaturation and native-membrane crosslinking. Given the position of L73Q deep within TM3, a movement of this residue to the membrane interface would require either (i) coordinated and highly costly insertion of periplasmic loops into the bilayer, or (ii) transient global unfolding followed by refolding. We detect no misfolding of ASBT-L73Q. Therefore, the enhanced degradation of ASBT-L73Q provides strong evidence that polar residues can promote recognition while remaining embedded in the membrane.

(b) Substrate engagement depends on the architecture of the FtsH transmembrane domain. Extending the hydrophobic core of FtsH TM1, thereby reducing its polarity and hydrophobic mismatch, markedly diminished membrane substrate interaction and degradation, while leaving cytosolic functions intact. This indicates that the membrane-embedded features of FtsH contribute directly to substrate engagement, which is not easily reconciled with a model based solely on surface-level interactions.

(c) Parallel findings from other membrane QC systems support intramembrane recognition of polar residues. Recent work across multiple organisms points to recognition events occurring within the bilayer, including the following papers about degradative QC:

* Lipid bilayer thinning near a ubiquitin ligase selects ER membrane proteins for degradation (bioRxiv 2025, doi:10.1101/2025.10.31.685944)

* Dsc ubiquitin ligase complex identifies transmembrane degrons at the Golgi (Weyer et al., Nat Commun 2024)

These independent studies, along with references 22 – 25, 30-33, and 76 support the broader principle that intramembrane polarity serves as a degradation signal across QC systems.

2. We have explicitly acknowledged the likely heterogeneity of unfolded states and already discussed these dynamics throughout the manuscript.

We agree that misfolded membrane proteins do not adopt a single conformation and may sample dynamic and heterogeneous states. We explicitly acknowledge this in the manuscript and did not intend Figures 1b or 3d to represent a literal structural model.

To further clarify this, we revised the Fig. 1b legend to state: *“While the model is likely not conformationally accurate, it illustrates that ...”*

We also already discussed the dynamic nature of misfolded states in several places in the manuscript:

- **Figure 1b legend:**
*“Membrane-embedded polar residues in the individual TMs of MdtJ represent a **hypothetical** fully unpacked unfolded state. Many polar residues **may potentially** face the lipid upon unfolding.”*
- **Main text (lines 143–145):**
“the folded and assembled MdtJ has 4 TMs and exposes a hydrophobic surface to the membrane (Fig. 1a). In the misfolded state, these helices may fully or partially unpack from each other, potentially exposing polar residues to the membrane (Fig. 3c)”

- **Main text (lines 328–332):**
“Interestingly, the finding that many endogenous polar residues of MdtJ were dispensable for degradation suggests that the position of the polar residue plays a role in the recognition. While the mechanistic explanation for this positional effect is unclear, it is plausible that the different positions display variable levels of lipid exposure in the dynamic misfolded state⁷⁷.”
- **Main text (lines 343–345):**
“How the FtsH ATPase initiates pulling in the absence of long cytosolic regions remains unclear. One possibility is that parts of the substrate TMs may transiently unfold, forming “cryptic” loops that extend from the membrane, as recently suggested for certain soluble FtsH substrates^{64,65}”

Together, these statements already convey that we do not assume a single, static unfolded structure.

Figures 1b and 3c/d are intended as didactic schematics rather than structural models, included to help readers outside the membrane-protein folding field appreciate a key conceptual point: polar residues are common within TMs, typically buried in the folded state, and can become exposed to the lipid environment upon misfolding. This context is difficult to convey textually without a simple visual, and the schematics illustrate only the location of these residues within individual helices, not an actual unfolded geometry. Although the unfolded ensemble is not structurally defined, we believe such illustrations are important for orienting non-experts. The revised legend now makes their illustrative nature explicit.

3. The reviewer’s concerns about marginal hydrophobicity may overestimate the extent of TM destabilization

Unlike ASBT, we cannot exclude topological dynamics for MdtJ. However, the available data suggest that its TMs remain well inserted under the conditions of our experiments. First, the hydrophobicity values in Fig. S2c show that the biological ΔG of these helices is <2 kcal/mol, a range that predicts robust insertion according to Bernsel et al. 2008 (PNAS, doi:10.1073/pnas.0711151105), as noted in our legend. The Bernsel study found that the biological hydrophobicity scale is skewed, and many natural TMs with positive ΔG values nevertheless insert efficiently.

Second, experimental topology mapping supports a stable insertion: in our previous study (Fluman et al., 2017 PNAS, doi:10.1073/pnas.1706905114), AMS-accessibility assays showed that the periplasmic loops connecting TMs 3&4 remain periplasm-exposed even when MdtJ is expressed without its partner. These results relied on a Cys

introduced into this loop in position 96 of MdtJ (in our PNAS paper, Fig. 1b,c pair 1, note that the name MdtJ did not exist at the time, and the protein is identified only by its Uniprot Accession). We now also include analogous unpublished data for the loop between TMs 1&2 (Cys37), which exhibit the same behavior. For interpretation of the gel, see our PNAS paper.

These results do not exclude transient topological dynamics, but they position the periplasmic loops of orphan MdtJ in the periplasm, placing the TMs away from the cytosolic FtsH ATPase domain.

2. Related to the point above: Generally, in a given catalytic reaction, the apparent outcome (here, degradation) could be decomposed into K_m (related to the binding of a substrate) and k_{cat} (the barrier towards ATP hydrolysis, active denaturation of substrates, and membrane extraction). I have an impression that in this work, the interpretation of the polar residue effect on degradation seems to be biased towards "binding", ignoring their possible effects (polar/nonpolar mutations on both substrates and FtsH) on the barriers in the major catalytic events (ATP hydrolysis, active denaturation, and membrane extraction).

-We agree that FtsH-mediated degradation comprises multiple catalytic steps, including substrate engagement, ATP hydrolysis, denaturation, and membrane extraction. Our *in vivo* degradation assays therefore measure the integrated outcome of all these steps and do not separate individual kinetic contributions such as K_m or k_{cat} . Because we did not measure ATPase activity or extraction energetics directly, we cannot meaningfully evaluate how polar residues might influence these downstream events, and discussing them in detail would risk overinterpretation rather than reduce bias.

The only step we could assess independently was substrate interaction, using the interaction assay shown in Fig. 3i and Fig. 6h. These experiments show that introducing polar residues into MdtJ, or altering the architecture of FtsH TM1, reduces formation of

the FtsH–MdtJ complex. This demonstrates that polar residues affect at least the engagement phase, without excluding additional effects at later catalytic steps.

Accordingly, in the manuscript we describe our observations in terms of changes in **degradation**, which is the experimentally accessible endpoint in vivo. For example:

- *“Additional substitutions in position 73 showed that the polarity of the residue correlated with degradation.”* (lines **118–119**)
- *“Indeed, both mutations accelerated the degradation of monomeric MdtJ by FtsH (Fig. 3g).”* (line **162**)

We also acknowledge that polar residues might influence later steps in the degradation pathway, and we already discuss this in the manuscript. For example:

- *“These residues could act indirectly, for example by recruiting an adaptor protein.”* (lines **358**)
- *“Such mismatch could aid substrate degradation in several ways ... facilitating extraction of the protein from the membrane by lowering the energetic barrier for retrotranslocation.”* (lines **362–368**)

In the revised version, we now add an explicit statement noting that:

- *“Notably, polar residues may also facilitate degradation by accelerating the extraction of MPs from the membrane by FtsH.”* (lines **333–334**)

Together, these points show that we do not restrict our interpretation to substrate engagement, but rather present a balanced discussion consistent with what our experiments can and cannot resolve.

3. The effect of polar/nonpolar mutations on FtsH is largely not conclusive. Many of these mutations retain activity for degrading water-soluble proteins. However, the activity for membrane proteins displays complex behaviors, in which the sites of effective changes are located at the residues outside the hydrocarbon core. This result renders it hard concluding that the region where the substrate recognition occurs is the membrane core. Authors acknowledges this point. Exposed polar residues on the substrates require their binding partners on FtsH. But, the lack of unequivocal recognition sites within TM1 in the membrane weakens authors' conclusion.

-We agree that the mutational effects in FtsH are distributed and that no single residue functions as an essential recognition site. This is precisely why we propose an architectural model, rather than a residue-specific one, in which the organization of the

transmembrane region facilitates substrate recognition. Notably, while not in the *middle* of the hydrocarbon core, the FtsH residues K6, N7, and Q23 are within the hydrophobic phase of the membrane, where polar residues are typically not found. As the reviewer noted, we have tried to delineate clearly what is and is not demonstrated in the current work, and the revised text reflected this.

Several lines of evidence support an architectural model:

- **Consistency with FtsH's promiscuous nature.**

FtsH, like other membrane QC systems, must recognize a wide range of substrates to surveil the membrane proteome. The diversity of its targets and the heterogeneity of their misfolded states argue against a single, highly specific binding pocket. Even if a binding site could be identified for one substrate, there is no reason to expect that a different substrate would engage FtsH in the same manner. For some substrates, additional factors also contribute to recognition. For example, LapB recruits LpxC to FtsH for degradation but is not required for degradation of other substrates (PMID: 38635633). Thus, the absence of an “unequivocal recognition site within TM1 in the membrane (hydrocarbon core)” is consistent with the broad and adaptable nature of FtsH substrate engagement.

- **Conservation of the TM architecture.**

The short hydrophobic TM segment capped with polar residues is conserved in bacteria (Fig. 6b,c; Fig. S6a) and present in several eukaryotic homologues (Fig. S6b). This architecture is compatible with models in which FtsH perturbs or thins the local membrane environment, and FtsH has been suggested to possess scramblase activity, which aligns with the idea of membrane remodeling.

- **Increasing evidence that membrane thinning and mismatch are general principles in membrane-protein QC.**

Recent studies from other systems indicate that intramembrane cues, including local thinning and hydrophobic mismatch, are used to recruit misfolded membrane proteins for degradation:

– *The Dsc ubiquitin ligase complex identifies transmembrane degrons to degrade orphaned proteins at the Golgi* (Nat Commun, 2024; <https://doi.org/10.1038/s41467-024-53676-6>).

– *Lipid bilayer thinning near a ubiquitin ligase selects ER membrane proteins for degradation* (bioRxiv, 2025; <https://doi.org/10.1101/2025.10.31.685944>).

Together, these observations support the idea that global or architectural features of the transmembrane region—rather than single recognition residues—contribute to the ability of FtsH to identify and process a broad range of membrane protein substrates.

Minor concerns:

1. Line 81: it is unclear why ASBT is an atypical membrane protein.

-We thank the reviewer for bringing this to our attention. ASBT is not an atypical protein, but the polar residue mutant is atypical. We changed the phrasing for clarification. The paragraph now reads:

“To examine whether lipid-facing polar residues can target a membrane protein for FtsH-mediated degradation, we utilized *Yersinia frederiksenii* ASBT, a bacterial homolog of the human apical sodium bile acid transporter. The structure of ASBT reveals a predominantly hydrophobic surface exposed to the lipid bilayer⁵⁷ (Fig. 2a left). We engineered it to present polar glutamine at position 73, in the middle of TM3, replacing a lipid-facing leucine (Fig. 2a, right). TM3 was selected since it is the most hydrophobic TM in ASBT, and is expected to robustly insert into the membrane upon mutation, avoiding misfolding due to problems in TM insertion (Fig. S1a). The mutated ASBT would thus atypically present a polar residue to the lipid bilayer, despite being folded.”

2. Fig. 5c: Although H147Y is a catalytically inactive mutant (probably, ATPase activity), this mutant seems to show the terminal cleavage activity (the reduced band intensity after 90 min). Some explanation seems to be needed.

-Indeed, the H417Y mutant appears unstable, but this does not reflect C-terminal self-cleavage. H417Y migrates as a higher band with anti-FtsH antibodies and is stably detected with antibodies against the C-terminal HA tag, both consistent with the absence of self-cleavage. This mutant disrupts the Zn²⁺-binding domain and is well documented in the literature (Westphal, 2012, PMID: 23091052; Arends, 2016, PMID: 27766750). The anti-FtsH signal indicates that H417Y undergoes partial degradation in cells, likely due to reduced stability, but we did not pursue this further because the underlying mechanism is not relevant to the question addressed here. The purpose of the mutant is simply to show that when FtsH is catalytically inactive, GFP-MdtJ is not degraded and its C-terminus remains intact.

3. Fig. 5c and lines 253-255: LacY TM1/2-FtsH chimera does not degrade MdtJ. However, Akiyama/Ito's work (2000, EMBO J; 2001, Biochemistry) has shown that the same chimera can degrade SecY. So, there should be a caution in generalizing the result.

-We agree that the LacY-FtsH chimera shows substrate-dependent behavior, and this should be taken into account. We already noted in the manuscript (Lines 237-241) that

although the *E. coli* FtsH TMs appear dispensable for some substrates, such as SecY, we speculated that “ FtsH may use distinct modes of recognition for different MP substrates”. To avoid ambiguity, we revised the text to clarify this further.

Specifically, Akiyama and Ito showed that the LacY–FtsH chimera can degrade SecY and σ^{32} , yet it does not fully complement an *ftsH* deletion strain, indicating that it cannot support all FtsH functions (Akiyama & Ito, 2000). We now add this point explicitly, in line 254 “*This may explain why the LacY–FtsH chimera could not fully complement an ftsH deletion, despite retaining activity against several substrates*47.”

Reviewer #3 (Remarks to the Author):

The authors have sufficiently addressed all my previous comments, and I applaud them on a much-improved manuscript. The additional data enhance the mechanistic insights and significance of their findings. All experiments are elegantly designed and well-controlled. I therefore recommend speedy publication of this study. I only have a few minor comments:

-We thank the reviewer for their kind words and constructive feedback, which helped improve the manuscript.

Minor comments:

In Fig. 2b the data points seem too big for the size of the panel.

-We fixed this now. Notably, this is a line, not a series of data points.

Panels in Fig. 2e/2f/3e/S3b and many more: Why are the connecting lines between data points squiggly?

-This was caused by the resolution of the PDF. Fixed.

Delete ‘where’ in: ‘These results confirm that the tertiary structure of ASBT-L73Q, especially related to TM3,4, and 9, where is maintained in the native membrane.’

-Thank you for noticing. It is deleted.

Fig. S2d: The legend should say color coding as in Fig. 3c

-Fixed.

Fig 3h+3i: Could the authors please provide a reference for their MdtJ-FtsH interaction assay based on the membrane mixing approach? Is this a standard assay in the E. coli field? Could the authors explain in more detail in the methods section how this works? Do the freeze-thaw cycles lead to membrane fusion, thus allowing the proteins to interact before detergent solubilization?

-Indeed, freeze-thaw-induced fusion is a standard method for mixing membrane vesicles, and we now clarify this in the Methods section. We added the following statement to lines 559–560:

“The mixture was subjected to three freeze-thaw cycles using liquid nitrogen to promote membrane vesicle fusion and interaction of embedded proteins.”

This step is required because the protease and substrate were expressed in separate cultures to prevent premature degradation before mixing. The interaction assay itself was developed in this study, and to our knowledge no prior reference describes this exact implementation.

Fig. 4g needs color coding of the different MdtJ variants

-Fixed.

Fig. 6b introducing residue numbering for E. coli FtsH TM1 would help with navigating the mutational data in 6d+e

-Fixed.

Reviewer #4 (Remarks to the Author):

The revised manuscript by Chai-Danino et al. extends our understanding of how misfolded/mis-assembled membrane proteins are recognized in the lipid bilayer, with a particular focus on the often understudied bacterial system. As such, it will be of interest to the field and in its revised form seems appropriate for publication. The authors may want to mention a recent study in line with their ideas (<https://www.biorxiv.org/content/10.1101/2025.10.31.685944v1>) in the final revision of their work.

-We thank the reviewer for the favorable review and for pointing out this recent work, which appeared after we submitted our manuscript. We have now incorporated it into

the relevant places in the Introduction and Discussion. Accordingly, we also modified line 325 to now include the ER E3 ligase: “and ER and Golgi-localized E3 ligases^{23–25,33}”